



# Evaluation of the dust-dominated total AOD extracted from the PMAp satellite Climate Data Record

Anu-Maija Sundström[1], Marie Doutriaux-Boucher[2], Soheila Jafariserajehlou[2], Dominika Leskow-Czyzewska[2], Simone Mantovani[3], Noemi Fazzini[3], Bertrand Fougnie[2], and Federico Fierli[2]

[1]Space and Earth Observation Centre, Finnish Meteorological Institute, Helsinki, FI-00560, Finland
[2]EUMETSAT, Darmstadt, DE-64295, Germany
[3]MEEO, Ferrara, I-44121, Italy

*Correspondence to*: Anu-Maija Sundström (anu-maija.sundstrom@fmi.fi)

**Abstract.** The Polar Multi-Sensor Aerosol optical properties product (PMAp) provides global Aerosol Optical Depth (AOD) observations that are retrieved using a combination of measurements from instruments onboard the Metop satellites, including the Global Ozone Monitoring Experiment-2 (GOME-2), the Infrared Atmospheric Sounding Interferometer (IASI), and the Advanced Very High Resolution Radiometer (AVHRR). The PMAp Climate Data Record (CDR), published in 2022, comprises data from the Metop-A and Metop-B satellites covering the period from 2007 to 2019. The PMAp also includes classification for selected aerosol types, including dust. Based on the classification, a dust-dominated total AOD can be extracted. The focus of this work is to assess the dust aerosols in the PMAp CDRs, by analysing the spatio-temporal occurrence of dust and aerosol classification reliability, as well as by carrying out dust-dominated total AOD validation against AErosol RObotic NETwork (AERONET) observations. Our results show that the occurrence and classification of PMAp dust-dominated AOD agrees well with AERONET metrics. For PMAp dust-dominated total AODs, moderate to strong correlations with AERONET (0.45–0.8) are observed, while mean biases exhibit relatively high variability. The root-mean-square errors (RMSEs) typically represent 50–80% of the mean AERONET AOD conditions. As most of the comparisons here occur at relatively high AOD levels over bright land surfaces, where measurement uncertainties and variability are inherently greater, this is somewhat expected. The results also bring up certain challenges, e.g. PMAp AOD overestimation at Central Asian AERONET stations or occasional occurrences of dust-dominated total AODs that appeared as clear outliers in AERONET comparisons. Further investigation is needed to determine their underlying causes. On a larger spatial scale, The PMAp CDRs can capture the expected seasonal variation in dust-affected AODs, such as over the Saharan outflow area, but sampling density can vary across seasons, especially over land. Therefore, full AOD distributions, along with median and mean, should be analyzed to ensure accurate conclusions. Despite challenges, the PMAp CDRs show potential for monitoring global dust aerosol patterns.



## 1 Introduction

Mineral dust is an essential component of Earth's atmosphere and plays a significant role in various environmental processes. Dust aerosols can influence the Earth's energy balance by scattering and absorbing solar and terrestrial radiation. This can lead to changes in regional and global climate patterns, affecting precipitation, temperature, and weather systems (e.g. Li and Wang, 2022). Dust particles can also affect climate indirectly by acting as cloud condensation nuclei and/or ice nucleating particles (e.g. Karydis et al., 2017, Roesch et al., 2021), perturbing cloud properties and potentially changing precipitation patterns

(Ansmann et al., 2008, Rosenfeld et al., 2001). Long-range transported dust and its deposition on the ground can influence nutrient cycling and soil fertility in various ecosystems (Prospero, et al., 2020, Mahowald et al., 2010), impacting vegetation growth and biodiversity.

Atmospheric dust aerosols significantly impact air quality and human health. The life cycle of dust —from emission, transport,
to deposition— is strongly controlled by atmospheric dynamics and thermodynamics (Kok et al., 2012). Frequent dust outbreaks can trigger severe air quality episodes, not only near dust source regions (e.g., Gama et al., 2020; Gkikas et al., 2013), but also in distant areas due to long-range transport (Ramírez-Romero et al., 2020; Morales-Medina et al., 2024). Inhaling dust particles can lead to serious health risks, including respiratory and cardiovascular issues (e.g., Dominguez-Rodriguez et al., 2020; Khaniabadi et al., 2017). Additionally, dust storms greatly reduce visibility, posing safety risks
especially in the transportation sector.

The so-called 'dust belt,' stretching from the west coast of North Africa through the Middle East to Central and South Asia, contains the world's largest dust emission sources. North Africa, particularly the Sahara Desert, is the largest contributor, accounting for about half of global dust emissions, while Asian source regions contribute roughly 40% (Kok et al., 2021).
Other regions, including North America, Southern Africa, South America, and Australia, have a much smaller impact on the global dust load. Dust emissions are influenced by meteorological conditions like surface winds and precipitation, as well as soil properties (e.g., bare soil fraction). These emissions show seasonal cycles unique to each source region, with diurnal variations as well (Vandenbussche et al., 2020; Schepanski et al., 2017; Zhao et al., 2022).

Satellites play a crucial role in observing and monitoring dust aerosols, providing valuable data on their spatial and temporal distribution at local, regional, and global scales, both in near-real-time and over extended periods. An increasing number of satellite products are being developed specifically to observe dust, utilizing both visible and thermal infrared remote sensing techniques. One of the key satellite-based parameters for dust monitoring is the dust (aerosol) optical depth (D(A)OD), that can be linked specifically to the amount of dust particles in an atmospheric column. Satellite-based DAOD products based on
observations at the visible wavelength range include, for example, the ModIs Dust AeroSol (MIDAS) dataset by Gkikas et al. (2020) and the DAOD climatology by Voss and Evan (2020). The MIDAS dataset provides global, fine-resolution (0.1° ×



0.1°) dust optical depth data spanning a 15-year period from 2003 to 2017. It combines Aerosol Optical Depth (AOD) observations from the MODerate resolution Imaging Spectroradiometer (MODIS) onboard the Aqua satellite with dust fractions derived from the Modern-Era Retrospective analysis for Research and Applications (MERRA-2) model to define

DAOD. Similarly, the global DAOD climatology by Voss and Evan (2020) is based on observations from MODIS onboard Terra and Aqua satellites, along with the AVHRR instruments. The DAOD is derived using aerosol typing from the ground-based AERONET observations (Holben et al., 1998). The DAOD climatology covers the period from 1981 (ocean) and 2001 (land) to 2018.

Dedicated dust optical depth products are also available from the thermal infrared observations e.g. from the Infrared Atmospheric Sounding Interferometer (IASI) onboard Metop-A, -B, and -C satellites. The advantages of using thermal infrared observations are the potential to carry out observations also in the absence of sunlight as well as the capability to differentiate aerosols by their composition. The IASI-based mineral dust datasets include e.g. The Mineral Aerosol Profiling from Infrared Radiances (MAPIR) that retrieves vertical dust concentration profiles from cloud-free observations (Callewaert et al., 2019),

Infrared Mineral Aerosol Retrieval Scheme (IMARS) that is a joint retrieval for mineral dust and ice clouds (Klüser et al., 2016), retrieval of dust characteristics by Laboratoire de Météorologie Dynamique (LMD, Peyridieu et al., 2013), and the neural network based DAOD ULB retrieval by Clarisse et al. (2019).

The Polar Multi-sensor Aerosol product (PMAp) is a global aerosol optical depth (AOD) dataset developed by EUMETSAT.

It combines simultaneous observations from multiple instruments onboard the Metop-A, -B, and -C satellites. The first operational version of PMAp was released in 2014 and has been continuously refined and improved since then (Grzegorski et al., 2022). In 2022, the first PMAp Climate Data Record (CDR) was released, covering observations from the Metop-A and -B satellites for the period 2007 to 2019 (Doutriaux-Boucher et al., 2022). In addition to providing total AOD for operational use, the PMAp retrieval provides also pixel-level information on selected aerosol types, including dust for non-operational

purposes. For preliminary aerosol typing and cloud correction the PMAp method uses observations from the IASI and the AVHRR instruments, while the actual AOD retrievals are carried out using the GOME-2 observations. The overarching goal of this work is to assess the dust aerosols in the PMAp CDRs from Metop-A and Metop-B. The primary goal of PMAp CDRs is to provide global, consistent observations on total AOD.It is of high interest to assess the applicability of PMAp CDR for global dust variability. The advantage of PMAp CDR is the relatively long temporal coverage, as well as good spatial coverage

for the period when both Metop-A and -B observations can be merged. The analysis of dust aerosols in the PMAp CDR is divided into two parts. First, the occurrence of dust in PMAp CDR is evaluated, and second, the accuracy of the retrieved dust-dominated total AODs is studied. It should be noted that the PMAp dust-dominated total AOD is different from DAOD, as it can consist also of contribution from other aerosol types than dust, whereas DAOD is the optical depth from mineral dust particles only. Comparisons of PMAp dust-dominated AODs against ground-based AERONET observations are carried out at

major dust emission source areas, including Northern Africa, Middle East and Asia.



This article is organized as follows: Section 2 provides an overview of the data used in this study while the methods are outlined in Section 3. The overview of the CDR evaluation of dust occurrence and accuracy of retrieved dust-dominated total AODs are presented in Section 4. Section 5 introduces examples of the large-scale seasonal variation of dust defined from the PMAp

CDRs. Section 6 summarizes the main conclusions of this work.

## 2 Data Description

### 2.1 Polar Multi-Sensor Aerosol product PMAp

### 2.1.1 PMAp retrieval approach

PMAp is a global satellite aerosol dataset developed by EUMETSAT. It provides aerosol optical depth (AOD) at 550 nm

along with an estimate of the dominant aerosol type. The first operational version of PMAp, released in 2014, included observations over the ocean only. In 2016, land retrievals were incorporated into the Near Real Time (NRT) system, and the dataset is now assimilated by the Copernicus Atmosphere Monitoring Service (CAMS; Garrigues et al., 2022). The PMAp retrieval is based on a multi-instrument approach, integrating simultaneous measurements from the GOME-2 Fundamental Data Record (FDR) Level 1C (http://doi.org/10.15770/EUM_SEC_CLM_0039), IASI FDR

(http://doi.org/10.15770/EUM_SEC_CLM_0014) + Level-1C NRT, and AVHRR Level 1B NRT data. The aerosol retrieval uses the GOME-2 Polarization Measurement Devices (PMDs) as the target footprint, with collocated measurements from AVHRR and IASI. The primary inputs for PMAp aerosol retrievals are GOME-2 PMD-P reflectances and Stokes fractions (Q/I). The footprint sizes for GOME-2 PMD measurements are 5 km x 40 km for Metop-A and 10 km x 40 km for Metop-B. Both Metop-A and Metop-B have an equatorial crossing time of 9:30 UTC.


The PMAp aerosol retrieval algorithm is composed of three successive steps that are described in detail in Grzegorski et al. (2022). The first step of the algorithm contains cloud identification and correction using collocated AVHRR observations, discrimination between aerosols and clouds as well as preliminary classification of aerosols, including desert dust detection. The desert dust detection is based on IASI observations and method developed by Clarisse et al. (2013, 2019), where the so-

called dust index is calculated from the IASI thermal infrared spectra. For volcanic ash/thick dust detection additional tests are carried out that are based on AVHRR and IASI brightness temperatures as well as ultraviolet absorbing aerosol index from GOME-2. The PMAp aerosol retrieval scheme can retrieve AODs also for partly cloudy footprints, if the cloud fraction is not too large and reflectances can be corrected for cloud contamination.

After the first step a set of candidates for the pixel-level aerosol types is obtained. The second step includes the retrieval of estimated AOD using unpolarised GOME-2 reflectances and a Look-Up-Table (LUT) based approach to optimise the



computational time. A set of AOD estimates is retrieved for each candidate aerosol model. The third retrieval step executes the actual minimization procedure used to find the best fit between the measured and modelled reflectances and Stokes fractions. Based on the best model, for each aerosol pixel, the aerosol type is set and the final AOD from the AODs retrieved

at the second step is selected accordingly. In PMAp the aerosol types are stored under the "aerosol_class" variable, that are: ocean fine mode, ocean coarse mode, thick biomass burning, desert dust, volcanic ash/thick dust, volcanic ash with SO2, aerosol contaminated cloud, ash contaminated cloud, and unclassified.

The PMAp aerosol retrieval scheme uses LUTs where four of the nine aerosol models used in the forward model represent

dust (Grzegorski et al., 2022, Hasekamp et al., 2008).  Each model is described with a bi-modal size distribution having an effective radius of 1.6 μm for the coarse mode and 0.1 μm for the fine mode, with a fixed fraction of large particle contribution. The four dust aerosol models differ by their absorption properties, and their assumed vertical distribution. Three of the dust models, two weakly and one moderately absorbing are used only over ocean retrievals, whereas a strongly absorbing model is used over land and ocean retrievals. The moderately absorbing dust model represents an elevated aerosol layer between 4 and

6 km, whereas for all other models aerosols are assumed to be within 0-2 km. For each aerosol model the optical properties have been calculated using the Mie-theory, i.e. assuming that particles are spherical (Hasekamp et al., 2008). Regardless of the aerosol type, the AOD maximum is set to 4.0 in the retrieval. If the resulting AOD after the minimization process is negative, it is set to 0.

The surface reflectance strongly affects the AOD retrievals at the visible wavelengths. Dust-dominated areas over land are typically associated with high surface reflectivity, which is known to be very challenging for AOD retrievals at the visible wavelength range (e.g. Ji et al., 2024, Huang et al., 2021).  In the PMAp retrieval the land surface reflectivity is represented using the GOME-2 Lambertian-Equivalent reflectance database (GLER, Tilstra et al., 2017).

**2.1.2 PMAp Climate Data Record**

The first PMAp CDR was released by EUMETSAT in September 2022 in the context of the Copernicus Climate Change Service (C3S). The PMAp CDR provides a consistent aerosol dataset over 13 years (2007-2019) of global aerosol properties consisting of observations from instruments onboard Metop-A and Metop-B satellites. The Metop-A PMAp CDR extends

from 1st July 2007 until 29th January 2018, and Metop-B from 20th February 2013 until 31st August 2019. The PMAp CDR is created using version 2.2.3 of the PMAp algorithm, which is aligned with the current NRT operational product in terms of scientific developments. The PMAp CDR data is accessible at http://doi.org/10.15770/EUM_SEC_CLM_0053 and is provided as one file per Metop orbit in NetCDF format.



The PMAp total AOD CDR validation overview  is presented in the EUMETSAT product validation report of the PMAp CDR
(PMAp, Release 1) by Doutriaux-Boucher et al. (2022). The report summarises that overall, the AOD CDRs are consistent
between Metop-A and Metop-B and that a good agreement against AERONET is obtained both over land and ocean stations.
The report also highlights challenges in retrieving AODs over bright surfaces, particularly in desert regions. According to
Doutriaux-Boucher et al. (2022), the largest discrepancies compared to AERONET occur in the subtropics, where biases

between PMAp total AOD and AERONET exceed 0.2, indicating a tendency for PMAp to overestimate the total AOD.
Correlations between PMAp and AERONET generally hover around 0.6, with an RMSE higher than 0.25. It is worth noting
that the results in the validation report differ somewhat from those presented in this study, mainly due to different filters used
here in data processing (Sect. 3.2). Figure 1 shows the temporal and spatial distributions of Metop-A and -B CDR total AODs.
Overall, a good consistency is found between the two CDRs for total AOD, both over land and ocean, in terms of temporal

and spatial variability on a global scale.

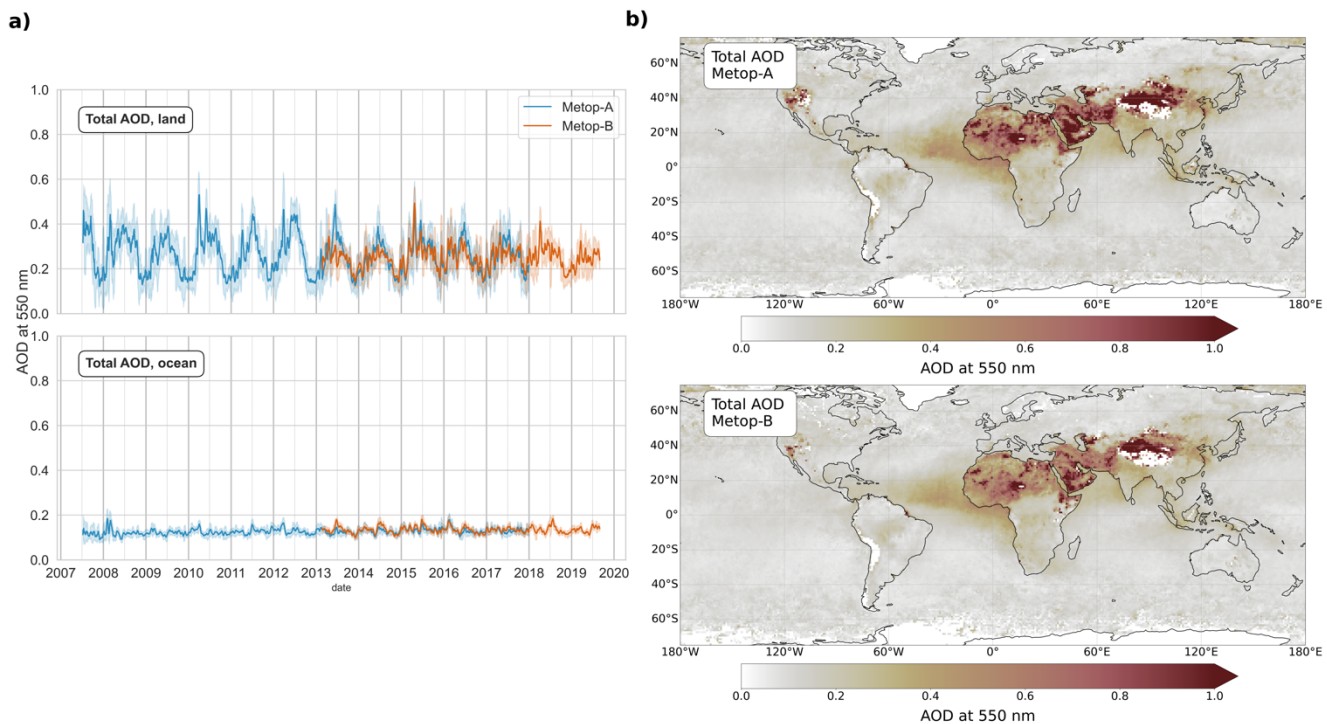

**Figure 1. a) Global mean of PMAp CDR for Metop-A (blue) and -B (orange) of total AOD over land (upper panel) and over ocean (lower panel). Shaded areas represent the AOD standard deviation.  b) An example of PMAp CDR annual mean of total AOD for**
**2015.**





**2.2 AERONET**


The Aerosol Robotic Network (AERONET) is a global network of Sun–sky photometers that have provided aerosol optical, microphysical, and radiative data for over 25 years. AERONET observations are key for validating aerosol satellite data, offering highly accurate AOD measurements (Holben et al., 1998, 2001). Data is freely available at https://aeronet.gsfc.nasa.gov/. The direct sun product provides AODs at wavelengths from 340 nm to 1020/1640 nm, as well

as the Ångström coefficient σ, while the inversion product (Dubovik et al., 2000) retrieves aerosol properties complex like refractive index and single scattering albedo (SSA). The AERONET direct sun data is provided at three quality levels: 1.0 (unscreened), 1.5 (cloud-screened), and 2.0 (quality-assured), with version 3 improving cloud screening and instrument quality control (Giles et al., 2019), while the inversion product is available as Level 1.5 and 2.0.

In this study, for the direct sun observations the highest quality, Level 2.0 data is considered. For inversion observations Level 1.5 data is selected, because the Level 2 observations are typically much less available than the direct sun. This is due e.g. to the requirement that cloud-free conditions are needed for the whole scan for the inversion, but also that there needs to be enough aerosols present for a high-quality (Level 2.0) retrieval. The inversion Level 1.5 product provides a flag that indicates those observations where Level 2.0 quality criteria are otherwise met except the requirement of AOD at 440 nm > 0.4. For this

study Level 1.5. inversion data was filtered according to this. As the interest from the inversion data is more on the spectral differences of SSA at 657 nm and 440 nm than the absolute SSA values, the use of Level 1.5 data can be considered justified.

The analysis of PMAp CDR dust-dominated AOD is carried out at five main dust dominated regions: Saharan Africa, Saharan outflow (Atlantic Ocean), Middle East, and Asia (Table 1). The AERONET stations used in this study have been selected

accordingly, considering also that the AERONET observations would extend over both Metop-A and -B CDR periods. In this work, PMAp CDRs are evaluated by collocating observations with selected AERONET stations and also considering broader regions to capture larger-scale variations. This approach allows for combining observations from multiple stations within the same area, providing a more comprehensive analysis. The study areas and the locations of the AERONET stations are shown in Figure 2.


**Table 1. List of AERONET stations used for PMAp dust validation. Data availability considers Level 2 direct sun observations that were available at the AERONET database in January 2024.**

| Station name | Lat, Lon | Area | Data coverage |
|---|---|---|---|
| Capo Verde | 16.73N, -22.93W | Sahara outflow, Atlantic | 2009-2023 |
| Tamanrasset INM | 22.79 N, 5.53E | Saharan Africa | 2006-2023 |



| | | | | |
|---|---|---|---|---|
| Banizoumbou | 13.55N, 2.67E | Saharan Africa | 1995-2022 | |
| IER Cinzana | 13.27N, -5.93W | Saharan Africa | 2004-2022 | 210 |
| KAUST Campus | 22.30N, 39.10E | Middle East | 2013-2023 | |
| Mezaira | 23.10N, 53.75E | Middle East | 2004-2019 | |
| Lahore | 38.55N, 68.85E | Asia 1 | 2010-2023 | |
| Karachi | 24.90N, 67.10E | Asia 1 | 2006-2023 | |
| Dalanganzad | 43.6N, 104.4E | Asia 2 | 1998-2022 | |
| Dushanbe | 38.55N, 68.85E | Asia 2 | 2010-2023 | |


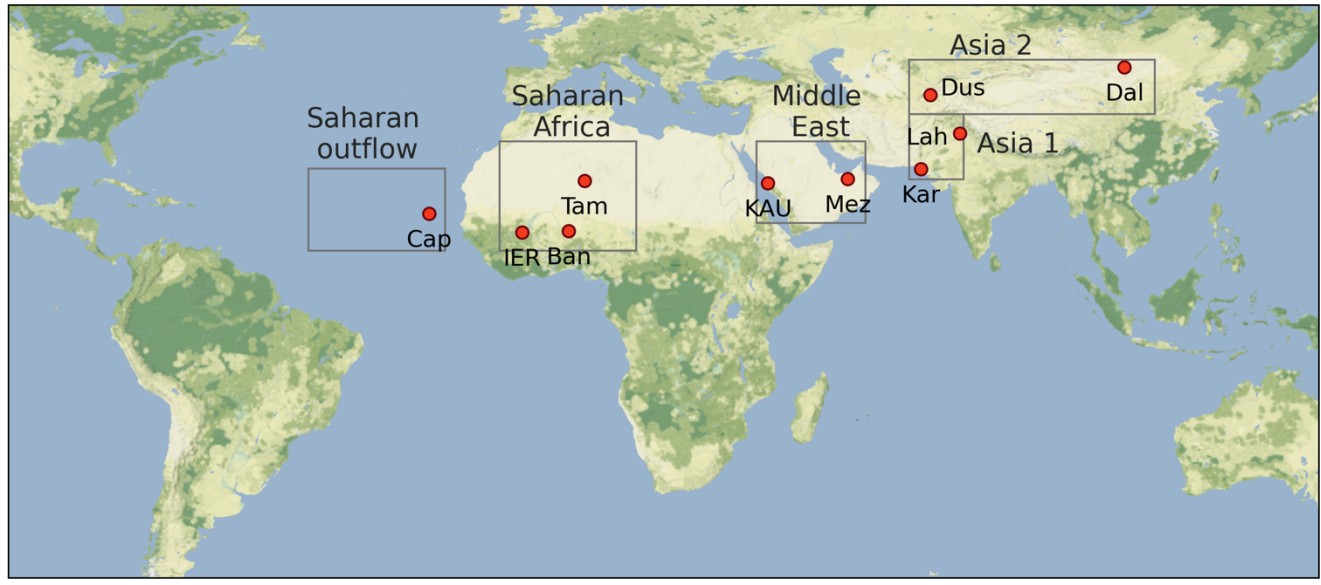

**Figure 2. Locations of the AERONET stations used at this study and definitions of the wider study areas.**

The three AERONET stations in Saharan Africa are heavily impacted by Saharan dust. Tamanrasset is in the Algerian Sahara highlands, Banizoumbou in Niger sits on a rural sandy plateau, and IER-Cinzana in Mali is at an agronomic research station. While there are no major industrial activities nearby, Banizoumbou and IER-Cinzana are affected by Sahel biomass burning, especially during the dry season (Cavalieri et al., 2010). The only site in an oceanic environment in this study is Capo Verde (Saharan outflow) located at Cape Verde archipelago in western Africa. Each year Cape Verde is affected by massive dust

plumes from the Sahara that can cross the Atlantic Ocean and affect air quality even in the Caribbean islands.





The two AERONET stations in the Middle East are on opposite sides of the Arabian Peninsula. KAUST Campus, in Saudi Arabia, is on the Red Sea coast, frequently impacted by dust from local deserts and northeastern Africa, with land/sea breezes influencing also the local dust distribution (Parajuli et al., 2020). Mezaira, in the UAE, experiences dust year-round from both local and distant sources (Nelli et al., 2021). The Asian region is divided into southern (Asia 1) and central (Asia 2) parts. Lahore and Karachi, Pakistan's largest cities, represent the southern region near the Thar desert, both heavily impacted also by anthropogenic emissions. Karachi is on the Arabian Sea coast, while Lahore is inland. The central Asia region includes Dushanbe, Tajikistan, located along major dust transport routes (e.g., from Aralkum, Taklamakan, and Karakum deserts) and affected by dust events mainly from April to November (Abdullaev and Sokolik, 2019). At the same time Dushanbe is also affected by anthropogenic pollution. Dalanzangad, in southern Mongolia near the Gobi Desert, is more rural than the other three stations but has seen rising human activity in recent years (Samman and Butt, 2023).

## 3. Methods

### 3.2 Definition of PMAp Dust-Dominated total AOD

PMAp retrieves AODs at 550 nm wavelength. Due to the available measurements in PMAp retrieval scheme, it is not possible to define the dust fraction (or AOD related to dust particles only) from a PMAp total AOD. This is because the dust retrieval is carried out using GOME-2 reflectances at visible wavelengths, and the dust contribution cannot be separated from other types of aerosols, even though the retrieval is carried out assuming a dust model. Therefore, alternative definitions need to be made. Here we define the dust-dominated total AOD, that is extracted from the PMAp CDRs by considering only those retrieved AODs, where the aerosol class in the footprint is set to "desert dust" (= class 3) or thick dust (=class 4 with ash flag=0). Hence, in this context the "dust-dominated" total AOD refers to pixels for which the assumed dominant aerosol type is dust, but it doesn't exclude the presence and contribution of other aerosol types to the observed AOD. In collocated and averaged PMAp data, the difference between the "total AOD" and "dust-dominated total AOD" is that total AOD is the average of all (filtered) observations, regardless of the aerosol type, while the dust-dominated total AOD is the average of only those CDR pixels where the aerosol class is set to "desert dust" and/or "thick dust". Hence, e.g. in spatially or temporally averaged data the total AOD and dust-dominated total AOD can differ in absolute values. For a valid "dust observation" in this study, not only must a dust-dominated pixel be identified, but the subsequent retrieval of AOD from GOME-2 must also be successful. Therefore, "occurrence of dust" here refers to those PMAp pixels where dust is detected and dust-dominated total AOD is retrieved.

It should be noted that the definition of PMAp dust-dominated total AOD is somewhat different from commonly used dust AOD (DAOD), that refers to the AOD that is attributable only to mineral dust particles while excluding the contribution from





other aerosol types. Therefore, the PMAp dust-dominated total AOD is not directly comparable in an absolute manner to those satellite-based products that report the DAOD, or at least these differences in definitions should be considered in the analysis.

## 3.2 Preliminary filtering and collocation of PMAp CDRs

For this study, both PMAp CDR datasets from Metop-A and Metop-B were pre-filtered prior to analysis. Pixels flagged for "snow and ice" were excluded. Additionally, as AOD values of 0.0 and 4.0 (upper limit of LUT) are not considered as fully physical in the retrieval process, only observations with $0.0 < AOD < 4.0$ were considered valid (Grzegorski et al., 2022). For Metop-A, only the narrow-swath observations were used, limiting the satellite scan angle to ±29° to maintain consistency over the entire observation period. This adjustment was necessary because in July 2013, the swath width of Metop-A was reduced from 1920 km to 960 km due to scan mirror degradation (Doutriaux-Boucher et al., 2022). Preliminary tests were also conducted to minimize potential cloud contamination in AOD values by comparing retrieved PMAp AODs with AERONET data as a function of PMAp cloud fraction and the difference between PMAp and AVHRR cloud fractions (see Appendix A). Based on these tests, the maximum PMAp cloud fraction was set at 0.4, and the minimum difference between PMAp and AVHRR at -0.6. Both CDRs were filtered accordingly. Since the original CDRs are provided only as orbit-based observations, the filtered PMAp CDRs were also gridded into a common 1° x 1° global grid on a daily scale for larger-scale studies. When calculating longer temporal averages from the gridded daily data, AODs were weighted in each grid cell based on their original pixel counts. Additionally, in time series, to account for varying grid cell areas towards the poles, weights were applied based on the cosine of the latitude. Details of the global gridding and defining dust-related AOD is discussed in more detail in Section 5.

For collocation of the PMAp data with AERONET stations similar approach as in Doutriaux-Boucher et al. (2022) and Grzegorski et al. (2022) was used, calculating a spatial mean of all those PMAp pixels with centre point falling within 30 km distance from the station. For temporal collocation two different criteria were used depending on the AERONET dataset. For direct sun AERONET observations a time window of ± 30 minutes of the satellite overpass is used. As the inversion data, including Single Scattering Albedo (SSA), is typically much sparser than the direct sun, a more relaxed temporal window of ± 90 minutes was considered to maintain the number of collocated observations at a level where meaningful comparisons can be carried out. As the AERONET nominal wavelengths do not include wavelength of 550 nm, the Ångström coefficient between 440 nm and 870 nm with AOD at 870 nm were used to derive AERONET AODs at 550 nm.



### 3.3 Evaluation approach

The evaluation of dust aerosols in the PMAp CDRs can be approached from two perspectives. First, assessing whether the occurrence of dust aligns spatially and temporally at larger scale with variability reported in the literature, as well as locally with AERONET observations. Secondly, examining the actual accuracy of the retrieved PMAp dust-dominated total AODs against AERONET.

While dust is expected to be the dominant component in the aerosol population at selected AERONET stations, some locations, especially in Asia and the Middle East are also regularly influenced by anthropogenic aerosols or sea salt in coastal areas. To focus specifically on dust cases when analysing local occurrence or dust-dominated total AOD accuracy, additional filtering of AERONET data was also used. Following the method outlined by Gkikas et al. (2021), a new subset of collocated data was created where filters were applied to AERONET observations based on Ångström coefficient $\alpha$ (440 nm-870 nm) and single

scattering albedo (SSA) at 675 nm and 440 nm. By limiting collocated AERONET observations to those where the $\alpha_{440\,nm-870\,nm} \leq 0.75$, it can be assumed that coarse-mode particles dominate the aerosol size distribution. The spectral dependence of AERONET single scattering albedo ($dSSA = SSA_{440nm} - SSA_{675nm}$) serves as a reliable indicator for identifying mineral dust particles, offering greater accuracy than absolute SSA values alone (e.g., Derimian et al., 2008). In this context, a negative dSSA indicates the presence of dust particles, while a positive dSSA indicates more black carbon type aerosols. Therefore,

applying the condition dSSA < 0 further discriminates coarse-mode-dominated cases as dust. However, as the additional AERONET filtering can significantly decrease the number of collocated observations at some of the stations, the validation of PMAp dust-dominated total AODs was performed with and without the additional AERONET filter.

### 4. Results and discussion

### 4.2 Occurrence of dust in the PMAp CDR

#### 4.2.1 Global variation of dust occurrence

Dust detection in the PMAp retrieval scheme is primarily driven by the IASI observations in the thermal infrared (dust index), which have enhanced sensitivity to coarse-mode dust particles (Clarisse et al., 2013). The PMAp dust detection scheme also incorporates additional tests based on AVHRR and GOME-2 data. Figure **3** presents the seasonal means of the fraction of dust-dominated AOD compared to total AOD for the entire Metop-B CDR period. The occurrence of dust-dominated AOD is

calculated by dividing the number of dust-dominated AOD observations by the total number of AOD observations. Each grid cell's observation count reflects the number of original footprints that pass the AOD filters. Dust occurrence patterns for Metop-A are available in Appendix A.





Globally, the highest PMAp dust occurrence fractions are observed along the dust belt, consistent with patterns seen in other
DAOD IASI products (e.g., Clarisse et al., 2019) and in dust datasets based on observations at visible wavelength range, such
as the LIdar climatology of vertical aerosol structure (LIVAS, Gkikas et al., 2020). The major dust seasons are clearly
represented in the PMAp CDRs. Dust transport from the Sahara to the Atlantic and Caribbean is prominent in the PMAp dust
occurrence for March-May (MAM) and especially June-August (JJA), aligning well with known seasonal dust transport cycles
from the Sahara (e.g., Kok et al., 2021). Also, frequent dust intrusions to the Western Mediterranean during JJA (Salvador et
al., 2014) are captured by PMAp observations, as well as the dust transport from the Gobi to the Pacific, peaking in spring
(MAM). Since dust occurrence is linked to the successful retrieval of AOD, seasonal dust patterns are somewhat influenced
by AOD observing conditions at the visible wavelength range. For instance, in Northern Asia, significantly fewer PMAp dust
observations are recorded in winter (DJF), partly due to snow cover in parts of the region, which contributes to the lower dust
occurrence percentages.


Results indicate that PMAp largely captures the global dust environment and seasonal cycles accurately. However, it appears
to miss some lesser dust source areas. According to Kok et al. (2021), minor dust sources in Australia, North America, South
Africa, and South America each contribute less than 5% to the global DAOD. While these minor dust sources in Namibia/South
Africa, Australia, and North America are visible in Figure **3**, the South American dust source e.g. in Patagonia is not similarly
detected. However, during DJF there is a weak outflow pattern over the South Atlantic Ocean, that could be related to Patagonia
dust. Additionally, some patterns in Figure **3** may be influenced by artefacts. For example, there are notable contrasts in PMAp
dust occurrences between land and ocean, particularly along the coasts of Africa and India, where dust occurrences
unexpectedly increase over the ocean. This anomaly could be attributed to a known bias in the IASI dust index, likely caused
by observations affected by temperature inversions (Clarisse et al., 2019).



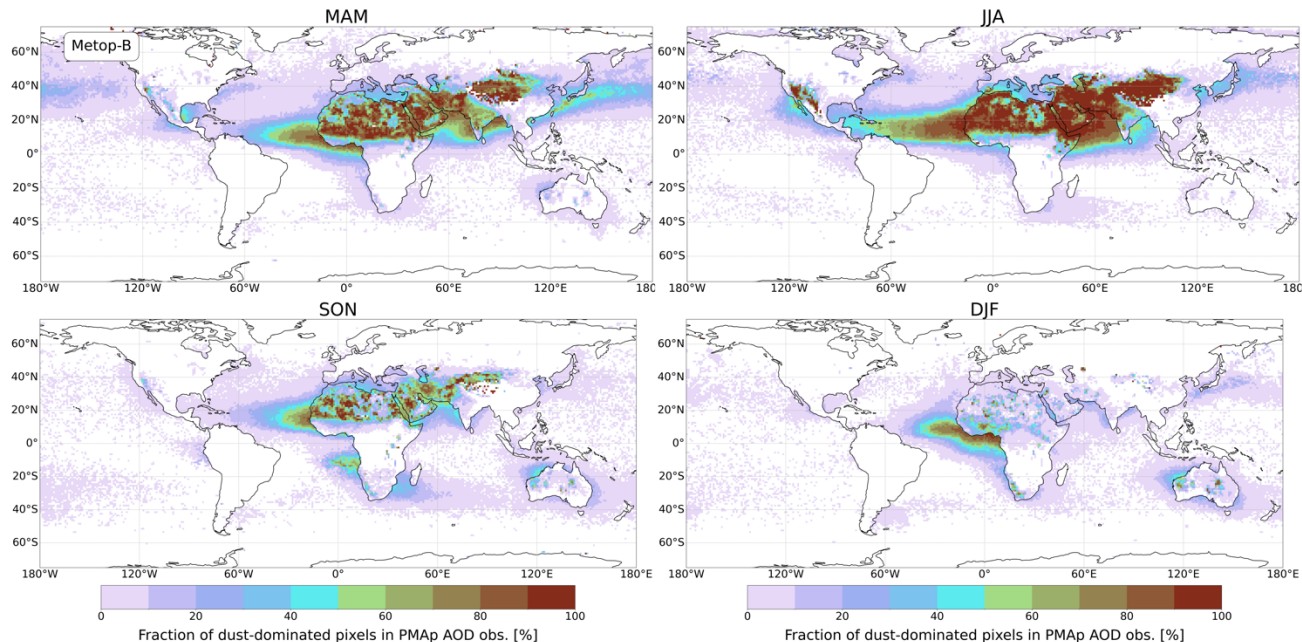

**Figure 3. Seasonal fraction of dust dominated pixels in PMAp AOD observations for Metop-B for the whole CDR observation period. Pixels with dust occurrence fraction < 0.5% have been excluded from the figure.**

### 4.2.2 Characterization of PMAp dust observations using AERONET

AERONET observations provide valuable insights for characterizing the aerosol optical properties of PMAp pixels dominated by dust. In this analysis, the absolute values of PMAp AODs are not considered. Instead, PMAp pixels, spatially collocated with AERONET, are categorized into two groups, "dust" and "other", based on the PMAp aerosol classification. Dust pixels are defined as those where the PMAp aerosol class is set as "desert dust" (=3) or "thick dust" (=4 with ash flag = 0), while the non-dust observations are categorized as "other". In fact, the results show that all PMAp pixels in the "other" category were assigned with an aerosol class of 15 in the PMAp scheme (both Metop-A and -B), which indicates that specific aerosol type for these pixels could not be identified within the available retrieval options. These two categories, "dust" and "other", are then compared against AERONET variables used for dust identification to explore how well the aerosol classification of PMAp is aligned with AERONET's dust classification.





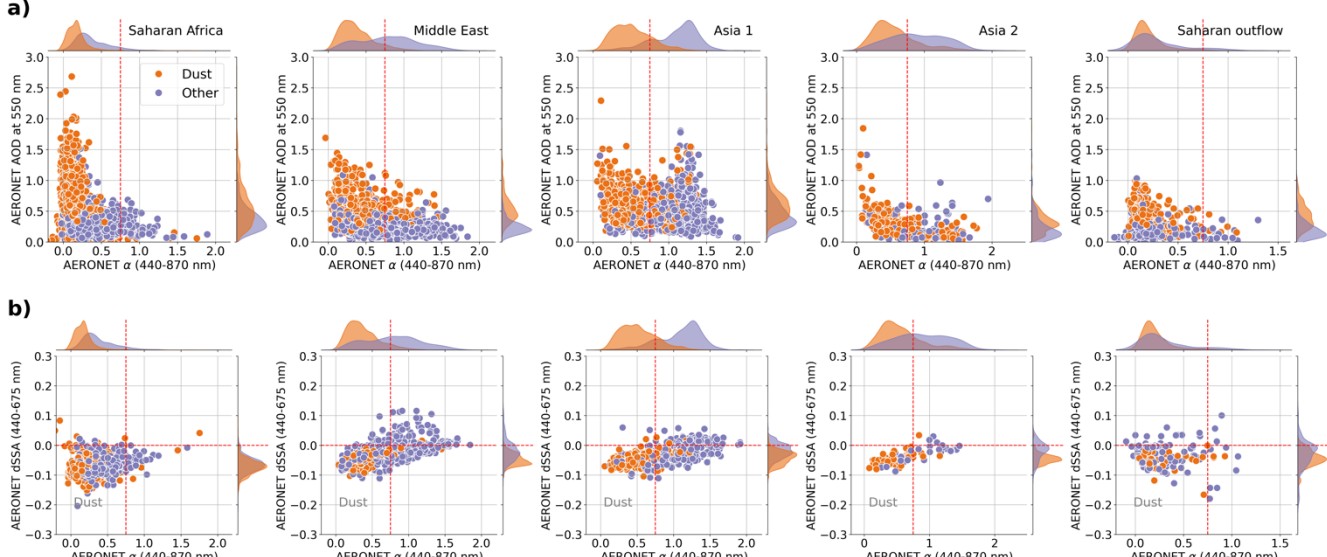

**Figure 4. Collocated PMAp observations from Metop-A and -B categorized as "dust" (orange) and "other type" (lilac) merged across AERONET stations. The PMAp observations are shown as points, whereas the density plots show the overall distribution of the corresponding variable, a) showing the relationship between AERONET AOD at 550 nm and the Ångström coefficient α to infer particle size vs. particle loading, and b) the spectral difference in SSA (dSSA) versus the α, i.e. aerosol composition vs. dominant particle size. Grey "Dust" label indicates the AERONET metrics used in this work to categorize AERONET dust cases from others. Red dashed lines indicate the thresholds for dust in AERONET metrics: α<0.75 and dSSA < 0.**

Figure 4 presents scatterplots of PMAp observations from Metop-A and Metop-B, classified as "dust" or "other" type aerosols, alongside AERONET data that are grouped by the study areas. The results in Figure 4a demonstrate that PMAp dust-labeled observations are predominantly characterized by coarse-mode particles ($\alpha_{440\,nm-870\,nm} < 0.75$) according to AERONET metrics. In contrast, PMAp "other" type observations exhibit more variability. In regions like Saharan Africa (Tamanrasset, Banizoumbou, IER Cinzana) and outflow (Capo Verde) areas, almost all PMAp observations are dominated by coarse-mode particles according to AERONET, regardless of the aerosol type. In the Saharan outflow area this is expected as the other dominant aerosol type in the area, sea salt, is also coarse mode dominated. However, in the other three regions, the "other" type PMAp observations tend to be centered around smaller particle sizes. A particularly distinct separation between PMAp "dust" and "other" regarding dominant aerosol size is observed in Asia 1 (Karachi, Lahore), where the two categories show a clear distinction in terms of AERONET α. This is expected, as both Karachi and Lahore are large cities and thus significantly influenced by anthropogenic emissions. Also, results show a relatively small, but systematic difference in AERONET AOD values between the PMAp "dust" and "other" observations. PMAp "dust" observations are generally more likely to be associated with higher AERONET AODs than the "other" type PMAp observations. The difference between AERONET AOD distributions for PMAp "dust" and "other" is especially clear e.g. in the Middle East and Saharan outflow areas. Overall, this



can indicate that at these AERONET locations high AOD episodes that PMAp observes are primarily caused by dust, and/or the PMAp dust identification works better in high AOD cases that at low AODs.

Figure 4b illustrates the distribution of PMAp "dust" and "other" observations in an AERONET α - dSSA space, the metrics
that is used in this work for AERONET observations to identify dust cases. Apart from a few outliers, all PMAp "dust" observations are associated with negative AERONET dSSA values, indicating chemical composition typical for dust. In Saharan Africa, the AERONET dSSA distributions for "dust" and "other" PMAp observations show little difference. However, in Asia 2 (Dushanbe, Dalanzadgad) and Saharan outflow regions, the distinction between the two PMAp categories is more apparent. Overall, this analysis suggests that the vast majority of PMAp "dust" observations correspond to dust, as
characterized by AERONET metrics. Nevertheless, there are several PMAp observations categorized as "other" that should also be classified as dust based on AERONET criteria.

## 4.3 PMAp AOD validation against AERONET

Two methods were employed to validate PMAp total and dust-dominated AODs against AERONET observations. The first approach involved a general validation of both total and dust-dominated AODs without filtering the AERONET data specifically for dust cases. This includes also a comparison of PMAp total AOD to assess whether significant differences emerge between the total PMAp AOD and those pixels that are specifically retrieved using the PMAp dust scheme. The analysis was carried out separately for Metop-A and -B at each AERONET station. The second approach narrow the validation
to dust-specific cases only, by comparing PMAp dust-dominated total AODs with AERONET data filtered for dust (using criteria of $\alpha_{440\ nm\ -\ 870\ nm} < 0.75$ and dSSA < 0).

### 4.3.1. Saharan Africa

At each station—Tamanrasset, IER Cinzana, and Banizoumbou—three comparisons between PMAp and AERONET AODs were conducted: 1) PMAp total AOD vs. AERONET AOD, 2) PMAp dust-dominated total AOD vs. AERONET AOD, and
3) PMAp dust-dominated total AOD vs. dust-filtered AERONET AOD. In Tamanrasset nearly all observations are categorized as dust by PMAp, and the number of collocated dust-dominated observations was the highest across all stations for both Metop-A and -B. Hence, there were only minimal differences between the total and dust-dominated total AOD correlations against (unfiltered) AERONET observations (Figure 5). For both CDRs, the correlations at Tamanrasset range from moderate to good, with mean PMAp biases (PMAp - AERONET) close to zero in the first two comparisons. The PMAp RMSEs are
approximately 0.25, indicating a moderate level of error compared to the mean AERONET AOD of about 0.45. When applying an additional dust filter to the AERONET data, the number of collocated observations decreases due to the limited availability





of AERONET SSA observations. In this third comparison, correlations slightly decrease at Tamanrasset, while bias and RMSE remain largely unchanged (Appendix A, Figure A2.1).

At Banizoumbou and IER Cinzana somewhat lower correlations and higher RMSEs are obtained for the dust-dominated total AODs than for total AODs, which suggest that the PMAp discrepancies are slightly more pronounced for dust pixels. Although the RMSEs at these two stations, especially for dust cases, are higher in absolute terms compared to Tamanrasset, they represent approximately 50% of the mean AERONET AODs, which is close to the level observed at Tamanrasset.  Figure 5 highlights a distinct pattern in PMAp dust-dominated AODs: at both IER Cinzana and Banizoumbou, there are very few PMAp

dust-dominated total AOD observations when AERONET AOD levels are low (<0.25). In contrast, this pattern is not observed for total AOD observations. In the third comparison with dust- filtered AERONET data at Banizoumbou the statistical metrics for PMAp dust-dominated AOD does not significantly change, at IER Cinzana an improvement is seen especially for Metop-B, with somewhat increasing correlation as well as decreasing mean bias and RMSE.





**Figure 5. Comparison of PMAp AODs against AERONET at three Saharan African stations: Tamanrasset (Algeria), IER Cinzana (Mali), and Banizoumbou (Nigeria). The color scale indicates number of observations, while black dashed line represents the 1:1 line.**

A typical feature of PMAp dust-dominated total AOD bias across each AERONET station is that it can exhibit considerable scatter throughout the AERONET AOD range, even at relatively low AOD levels, regardless of whether the comparison is carried out against "unfiltered" or dust-filtered AERONET AODs. Results differ somewhat between Tamanrasset and the two other stations, but a common feature is that the PMAp dust-dominated mean bias tend to be rather stable for AERONET AOD levels lower than about 0.5: at Tamanrasset close to zero and at IER Cinzana and Banizoumbou slightly negative (Appendix



A, Figure A2.2). At Tamanrasset for higher AERONET AODs, particularly above 1.0, the bias becomes increasingly negative while at at IER Cinzana and Banizoumbou no clear systematic pattern emerges. Across all three stations, weak seasonal variability in PMAp AOD bias is observed, corresponding with the West African monsoon cycle (Cuesta et al., 2008). During the rainy season (mainly summer months), the bias tends to be more negative, whereas in the dry season, the bias shifts toward more positive values.


Figure 6 shows an example of PMAp dust-dominated AOD dynamical variation as compared to AERONET total AOD at Tamanrasset between January 2013 and December 2017, demonstrating consistent patterns for both Metop-A and Metop-B. In Tamanrasset AERONET data reveals a clear seasonal variation, with lower AOD values in (Northern Hemisphere) winter and higher values in spring/summer, which is highly driven by dust activity in the area. As Figure 6 shows for both timeseries,

PMAp dust-dominated total AODs generally capture this seasonal pattern, even though there are overall a lower number of PMAp dust observations available during winter. For both timeseries there are also some apparent outliers detected, when PMAp dust-dominated AODs show very high values compared to AERONET. Figure 6 shows also a closer look on a dust season in 2015. Even though both PMAp timeseries lack observations at the highest AOD peak in July, for the most part both CDRs seem to agree with AERONET throughout the period.  However, Metop-B appears to show slightly higher occurrence

of extremely low, near-zero, dust-dominated total AOD values, which indicates an underestimation when compared to AERONET values. Similar pattern of very low dust-dominated total AOD values for Metop-B are also observed at the two other Saharan stations, IER Cinzana and Banizoumbou.

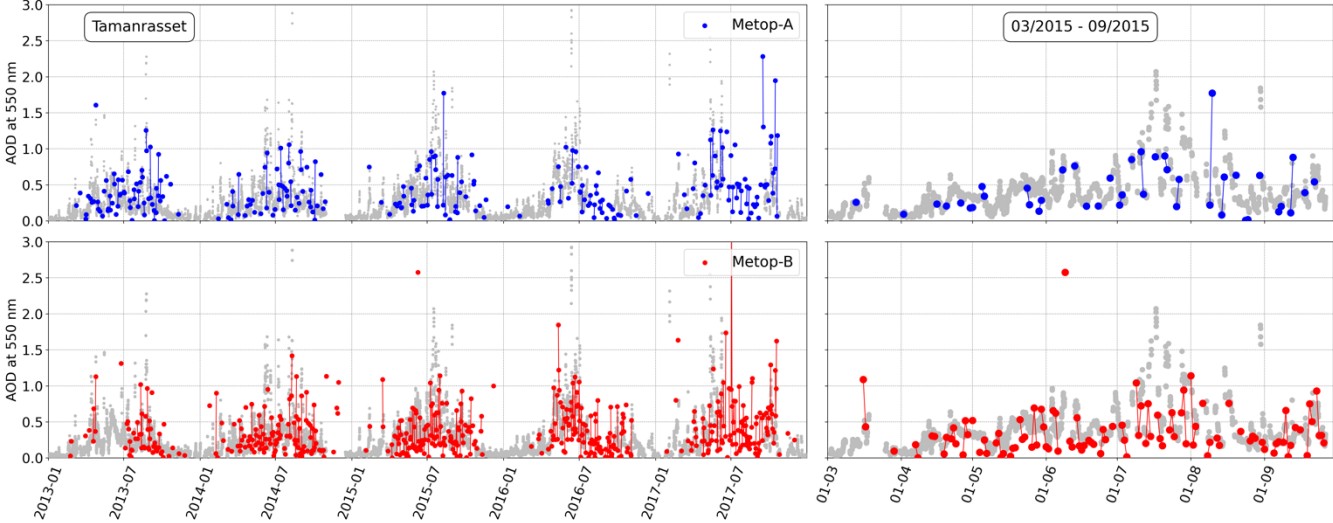

**Figure 6. Temporal variation of dust-dominated total AOD measured by PMAp for Metop-A (blue) and Metop-B (red). The left panel displays data from January 2013 to December 2017, while the right panel focuses on the period from March 2015 to September 2015 at Tamanrasset, alongside AERONET total AOD measurements (grey). Those PMAp points connected by a line are observations from two consecutive days.**





### 4.3.2 Middle East

A key difference between the KAUST Campus and Mezaira AERONET stations lies in their geographic settings: KAUST Campus is located on the coast, with nearly 30% of collocated PMAp observations taken over the ocean, while all observations at Mezaira are taken over land. This partially explains the differences particularly in the PMAp total AOD comparisons shown in Figure 7. Observations over the ocean are generally less affected by surface-related effects compared to those over bright land surfaces. At Mezaira, several instances show PMAp total AOD values near zero, despite AERONET showing significantly higher AODs. This pattern is similarly observed at IER Cinzana and Banizoumbou (Figure 5). In contrast, KAUST Campus exhibits fewer such discrepancies, with a generally better agreement between PMAp total AOD and AERONET, even at low AOD levels.

While the correlation between PMAp dust-dominated total AOD and AERONET AOD (with and without the dust filter) ranges from moderate to good (Metop-A: 0.5–0.84, Metop-B: 0.43–0.55), the high mean biases and RMSEs reveal significant discrepancies in absolute values (Figure 7, Appendix A, Figure A2.3). At both stations, the dust-dominated RMSE can exceed 70% of the corresponding mean AERONET AOD, highlighting a higher relative error compared to the observations in Saharan Africa. The PMAp dust-dominated total AOD bias shows also considerable variability across different AERONET AOD levels at both Middle Eastern stations (Appendix A, Figure A2.4), with minor differences observed between Metop-A and Metop-B. For both CDRs, the mean bias remains relatively stable at AOD levels below 0.5, being slightly positive for Metop-A and slightly negative for Metop-B. However, substantial scatter in bias values persists, even at relatively low AOD levels, particularly at KAUST Campus. Similar patterns are observed when comparing against dust-filtered AERONET AODs.



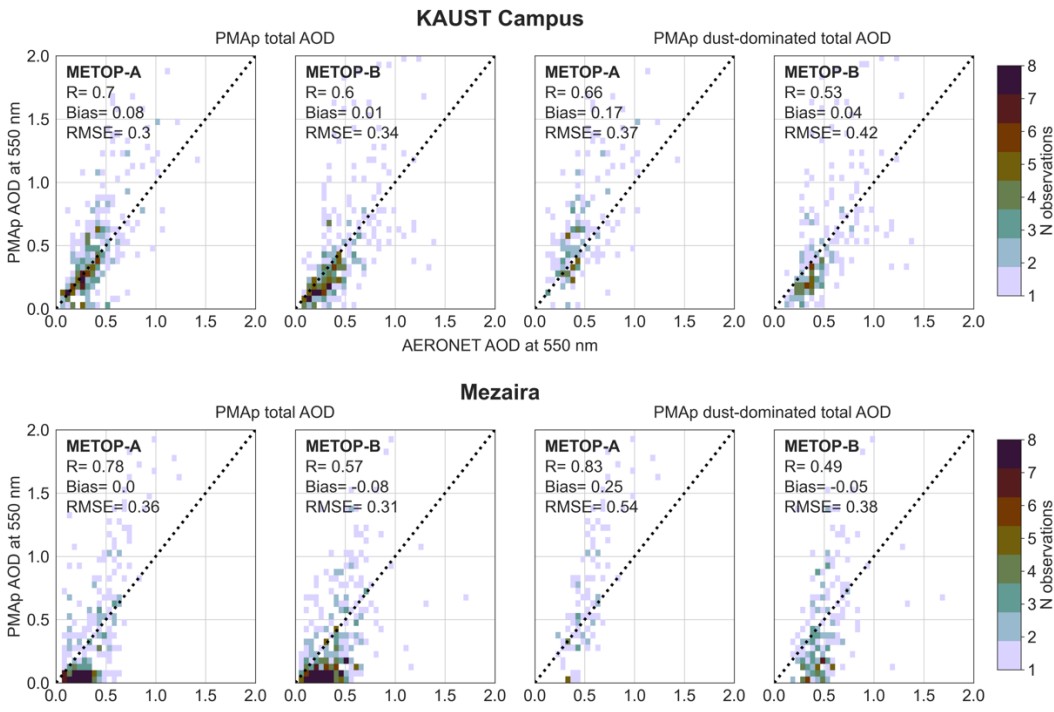

**Figure 7. Comparison of PMAp AODs against AERONET in KAUST Campus (Saudi Arabia) and Mezaira (Unites Arab Emirates). The color scale indicates number of observations, while black dashed line represents the 1:1 line.**

480

As shown in the time series for KAUST Campus (Figure 8), the seasonal variation of AOD is somewhat less distinct compared e.g. to Tamanrasset (Figure 6). Although PMAp dust-dominated data sampling is somewhat sparser at KAUST Campus, both (Metop -A and -B) PMAp dust-related time series generally capture the dynamic variations well. A notable period of increased dust activity between May and August 2015 (right panel, Figure 8) is successfully detected by both satellites. However, at the

485 same time occasional outliers in the PMAp dust-dominated AODs are observed for both satellites. While some of the higher PMAp dust-dominated AOD values are clearly associated with actual dust episodes, no single clear cause for these sporadic outliers in the PMAp CDRs could not been identified in this analysis.





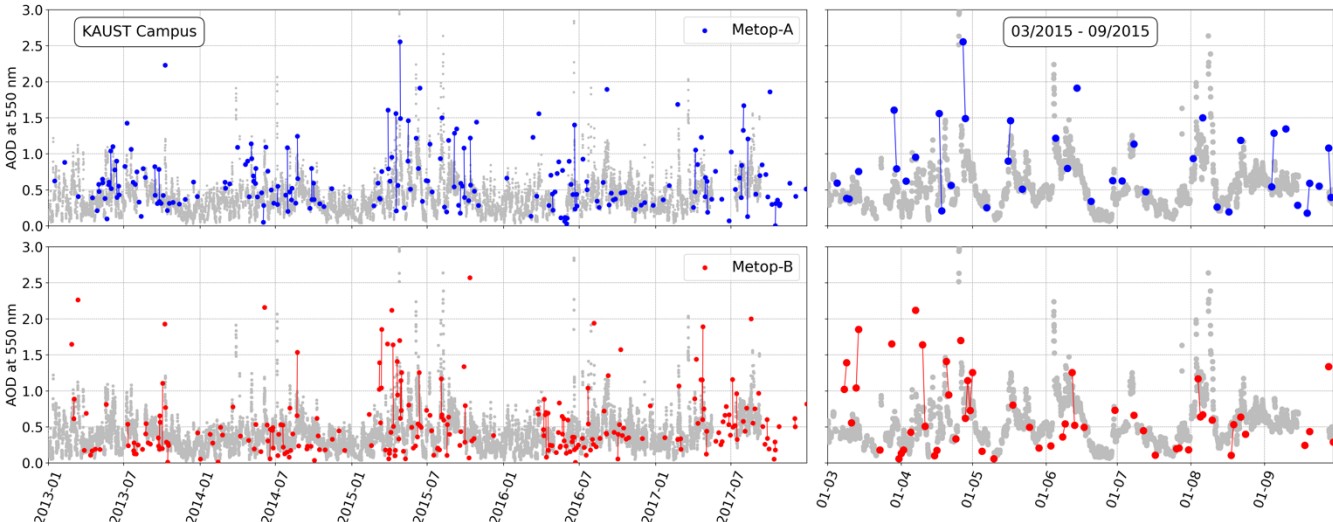

**Figure 8. Temporal variation of dust-dominated total AOD measured by PMAp for Metop-A (blue) and Metop-B (red) at Mezaira. The left panel displays data from January 2013 to December 2017, while the right panel focuses on the period from March 2015 to September 2015, alongside AERONET total AOD measurements (grey).**

### 4.3.3. Asia 1

The region consists of two AERONET stations located in major cities, Lahore and Karachi, where anthropogenic emissions play a significant role on the aerosol environment. Correlations with AERONET for both Metop-A and Metop-B, for total or dust-dominated total AODs are fairly similar, ranging from 0.6 to 0.8, except in Karachi, where Metop-B displays somewhat lower correlations for dust-dominated total AOD (Figure 9). The dust-dominated total AOD RMSEs are notably high, exceeding 50% of the mean AERONET AOD. Including a dust filter for the AERONET data appears to slightly improve the comparisons, particularly by reducing RMSEs, despite a decrease in the number of collocated observations. An exception is observed for Metop-B in Karachi, where RMSEs remain elevated at approximately 80% of the mean dust-filtered AERONET AOD (Appendix A, Figure A2.5). In other cases, RMSEs are generally reduced to 50% or less of the mean dust-filtered AERONET AOD, indicating a lower relative error.

At Lahore, the overall mean biases for all three comparisons remain negative for both Metop-A and Metop-B, whereas at Karachi, the mean biases are positive. Similar to the results observed at Saharan Africa and Middle East stations, the dust-dominated AOD bias shows considerable scatter, which does not appear to depend on AERONET AOD levels. Although the scatter is slightly reduced when dust-filtered AERONET observations are used, the overall characteristics and behaviour of the PMAp dust-dominated bias remain consistent across both instruments.






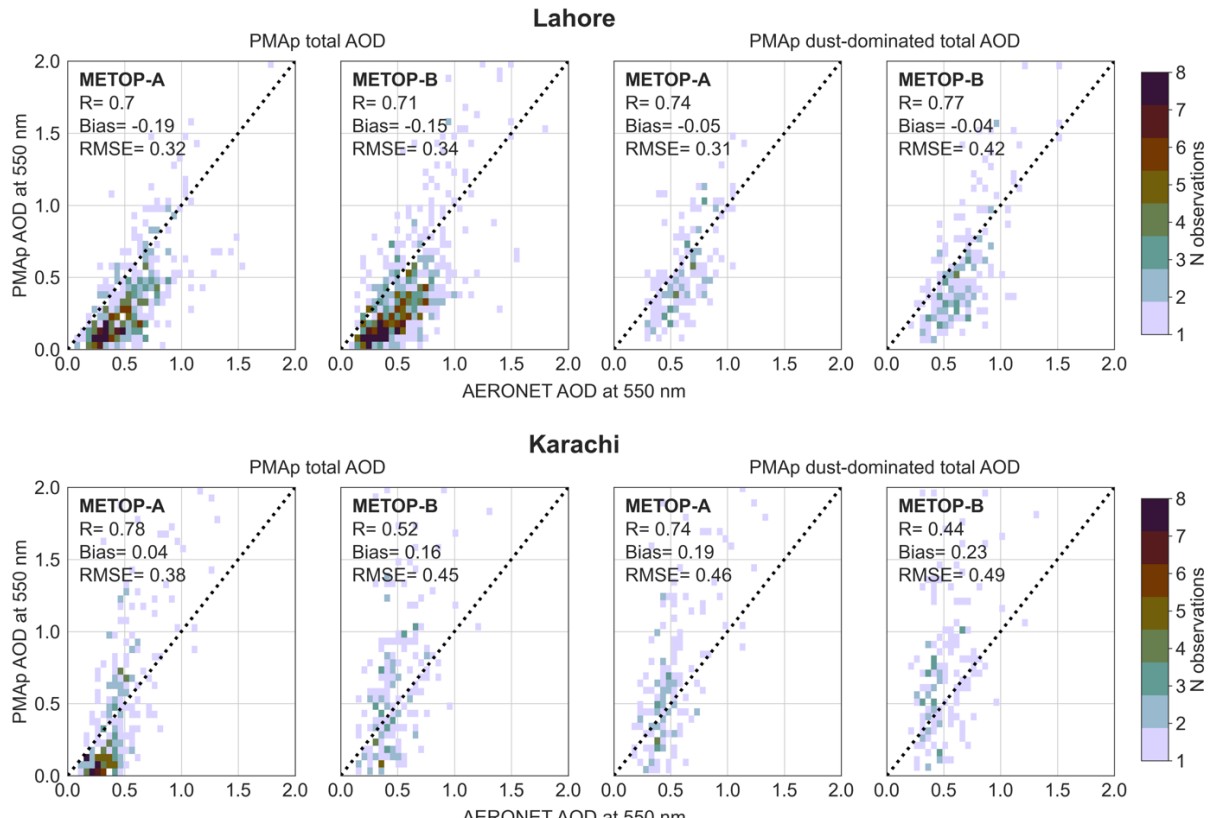

**Figure 9. Comparison of PMAp AODs against AERONET in Lahore and Karachi (Pakistan). The colour scale indicates number of observations, while black dashed line represents the 1:1 line.**

The AERONET time series in Lahore is significantly influenced by anthropogenic activity, which is reflected in Figure 10 as
a high variability in AOD. As shown in Figure 4 shows, the observed AOD peaks can result from both coarse- and fine-mode
aerosol events. While PMAp tends to underestimate AOD in Lahore and dust-dominated observations are limited, it still
captures the overall dynamic variations reasonably. The underestimation seems to be more pronounced at lower AOD levels,
while at elevated AOD levels, PMAp performs somewhat better in capturing the observed variability.



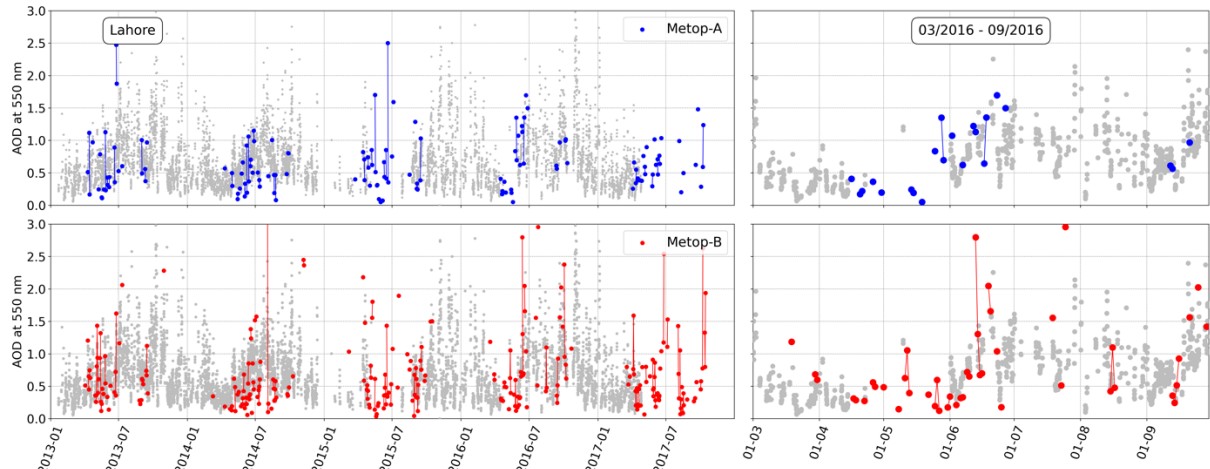


**Figure 10.** Temporal variation of dust-dominated total AOD measured by PMAp for Metop-A (blue) and Metop-B (red) at Lahore. The left panel displays data from January 2013 to December 2017, while the right panel focuses on the period from March 2016 to September 2016, alongside AERONET total AOD measurements (grey).

### 525 4.3.4. Asia 2

Dushanbe and Dalanzadgad, located in the northern part of Central/Eastern Asia, are also influenced by anthropogenic activity, though to a lesser extent than Lahore and Karachi. At both stations, ground-based AERONET AOD values remain relatively low (<0.5). However, PMAp total AOD and dust-dominated total AOD exhibit systematically high positive mean biases (0.21–

0.38) and substantial RMSEs (0.35–0.56, Figure 11). At Dushanbe, the biases and RMSEs are the highest among all stations in this study. It is also noted that the number of collocated PMAp observations for both Metop-A and Metop-B is the lowest in this area. The AERONET dust filter could only be applied at Dushanbe, as the remaining number of collocated observations at Dalanzadgad was insufficient. Regardless, the RMSEs for PMAp dust-dominated total AODs exceed the mean AERONET AOD (with and without the dust filter), highlighting a substantial level of error relative to the observed values at these stations.





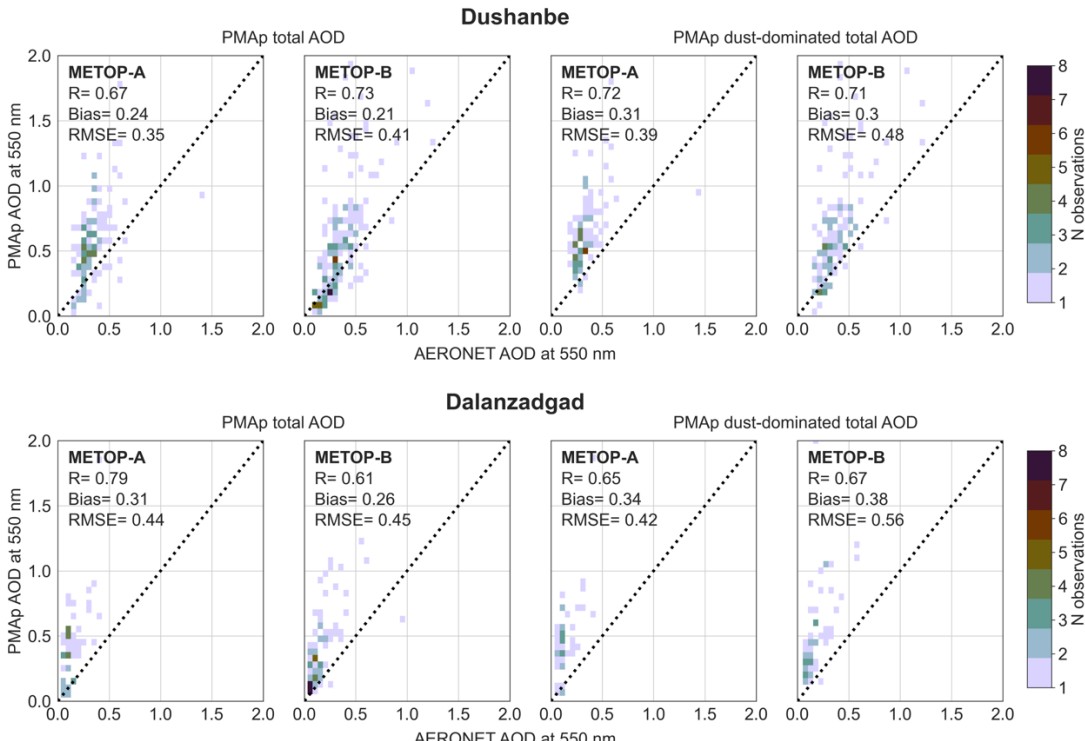


**Figure 11. Comparison of PMAp AODs against AERONET in Dushanbe (Tajikistan), and Dalanzadgad (Mongolia). The color scale indicates number of observations, while black dashed line represents the 1:1 line.**

Time series data from Dushanbe indicate that dust-dominated total AOD observations from PMAp are primarily recorded

between June and September, aligning with the season when AERONET also reports the highest AOD episodes (Figure 12). This pattern is consistent with documented dust activity in the region (Abdullaev and Sokolik, 2019), suggesting that dust likely plays a significant role in the most elevated AOD levels during summer and early autumn. Oveview of both Metop-A and -B timeseries in Dushanbe shows clearly the general PMAp overestimation, however a closer look at a shorter period in 2013 show that some of the dynamic variability at lower AOD levels is still captured.




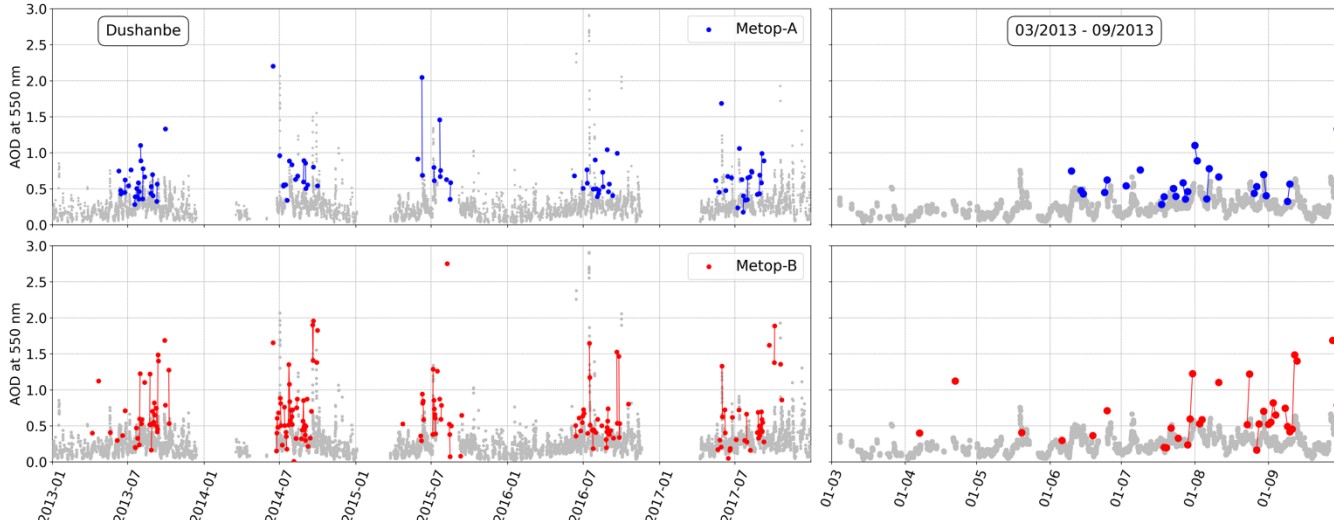

**Figure 12. Temporal variation of dust-dominated total AOD measured by PMAp for Metop-A (blue) and Metop-B (red) at Dushanbe The left panel displays data from January 2013 to December 2017, while the right panel focuses on the period from March 2013 to September 2013, alongside AERONET total AOD measurements (grey).**


### 4.3.5. Saharan outflow

Unlike other stations, all PMAp observations at Capo Verde are categorized as retrievals over water, which differs from the retrieval method over land. For ocean retrievals, if dust is detected, PMAp employs three distinct dust aerosol models
(compared to a single model over land) to achieve the best fit. In addition, for dust-classified pixels GOME-2 reflectances are not corrected for clouds but it is assumed that the scene contains only aerosols. When compared with AERONET data, PMAp data (both total and dust-dominated total AODs) show relatively low mean biases and RMSEs for both satellites (Figure 13). Adding the dust filter to the AERONET data improves the correlations and RMSEs. However, for Metop-B the mean bias increases in absolute values from -0.04 to -0.11 (Appendix A, Figure A2.10).


At Capo Verde, the overall scatter of the PMAp dust-dominated AOD bias is lower compared to other stations across all AERONET AOD levels (Appendix A, Figure A2.11). The mean dust-dominated AOD bias shows a slight decreasing trend as AERONET AOD values increase. Applying the dust filter to the AERONET data further reduces the scatter in the dust-dominated total AOD bias; however, a tendency for negative bias persists at AOD values greater than 0.5.



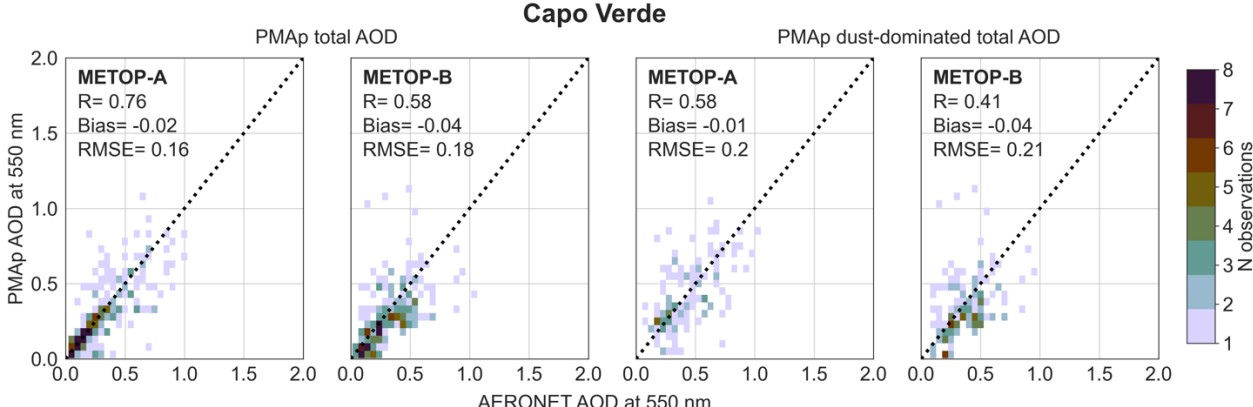

**Figure 13. Comparison of PMAp dust-dominated and total AODs against AERONET at Capo Verde for Metop-A and -B.**

At Capo Verde, the AERONET data shows a seasonal cycle similar to that observed in Tamanrasset, though less pronounced (Figure 14). Overall, PMAp's dust-dominated CDR captures this variability well. While a few PMAp outliers appear in the time series, both Metop-A and Metop-B exhibit fewer clear outliers compared to other stations considered in this study. This suggests that PMAp outliers could be specifically associated with dust retrievals over land.

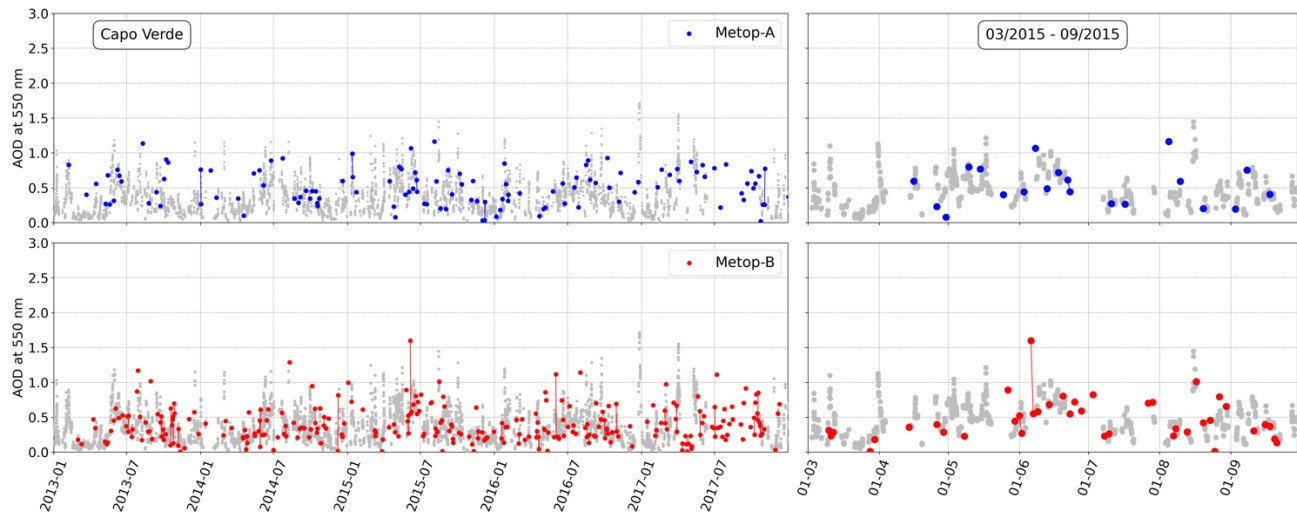

**Figure 14. Temporal variation of dust-dominated total AOD measured by PMAp for Metop-A (blue) and Metop-B (red) at Dushanbe The left panel displays data from January 2013 to December 2017, while the right panel focuses on the period from March 2015 to September 2015, alongside AERONET total AOD measurements (grey).**



## 4.4 Spatial overview of the AERONET validation and statistical metrics


Figure 15 illustrates the spatial distribution of statistical metrics for PMAp dust-dominated total AOD compared to AERONET AOD without dust filtering. As discussed in earlier sections, applying the dust filter to AERONET data did not significantly alter the results for PMAp dust-dominated AOD. Overall, the spatial overview reveals minimal systematic patterns in the metrics for either Metop-A or Metop-B. However, Central Asia (Asia 2) stands out with consistently high positive biases and elevated RMSEs for both instruments, highlighting discrepancies in PMAp data for this region. On the other hand e.g. over Africa and Saharan outflow regions the mean biases for dust-dominated total AODs are lower.

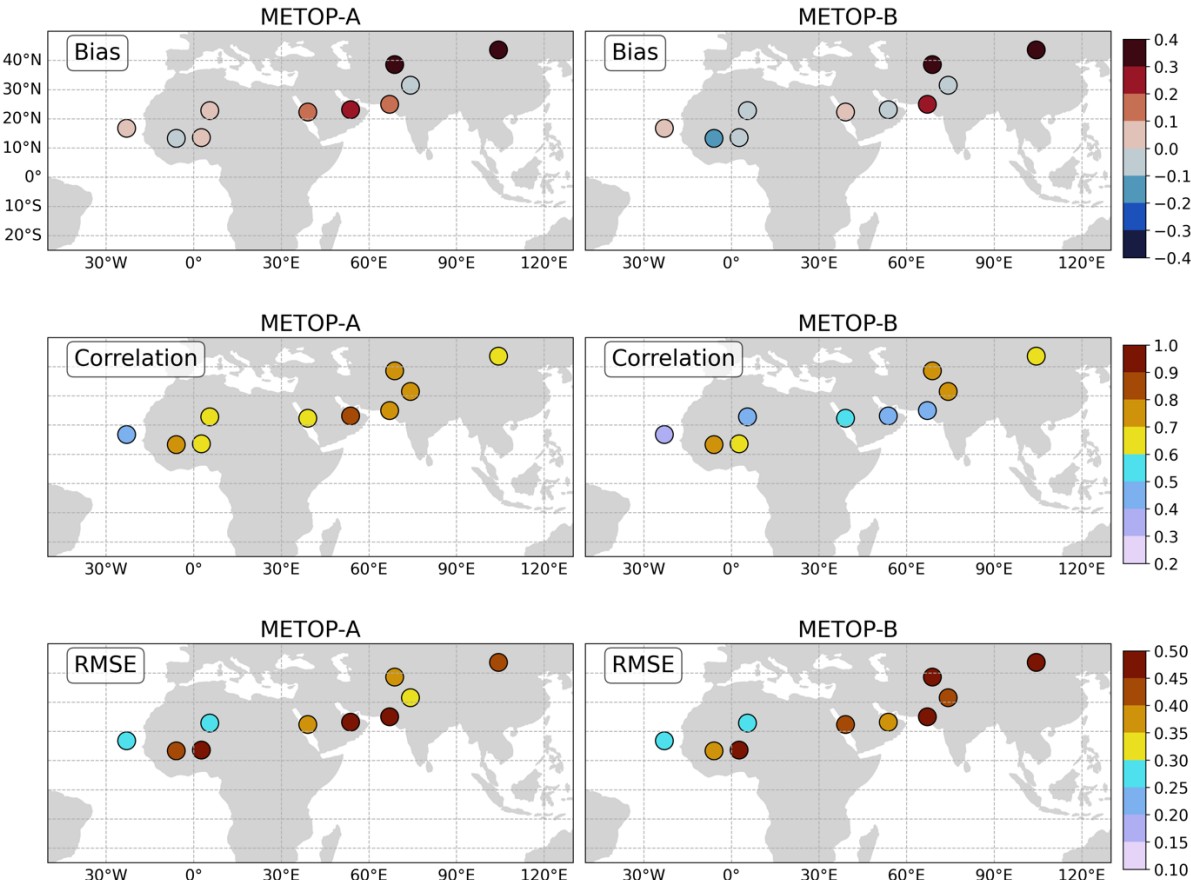

**Figure 15. Summary of the statistical metrics obtained for PMAp dust-dominated total AODs against AERONET total AOD.**



# 5. Global variation of PMAp dust-related AOD

Defining an AOD that represents "dust" for a globally gridded data using PMAp observations is not straightforward. As
discussed in earlier sections, the observations used in PMAp retrievals do not allow for the separation of dust from the total
aerosol signal. Therefore, the PMAp observations as such cannot be used to define DAOD (i.e. dust fraction from total AOD).
The PMAp pixels labeled as dust cannot be assumed to contain only dust particles, similarly, the non-dust pixels cannot be
considered entirely free of dust. Furthermore, as highlighted in Sect. 4.2.2 through AERONET characterization, a significant
number of PMAp pixels labelled as aerosol types other than dust are likely to be substantially influenced by dust. Consequently,
there are likely multiple approaches to defining "dust-related" or "dust-dominated" total AOD from the PMAp CDRs.

For the global scale analysis, the PMAp CDR data in this study is gridded so that each 1° x 1° grid cell with a dust observation
(i.e., sum of dust-dominated total AODs > 0) is labelled as a dust-affected grid cell. Further, the dust-related AOD for each
grid cell is then defined as the mean of all valid PMAp AODs, regardless of the aerosol type.


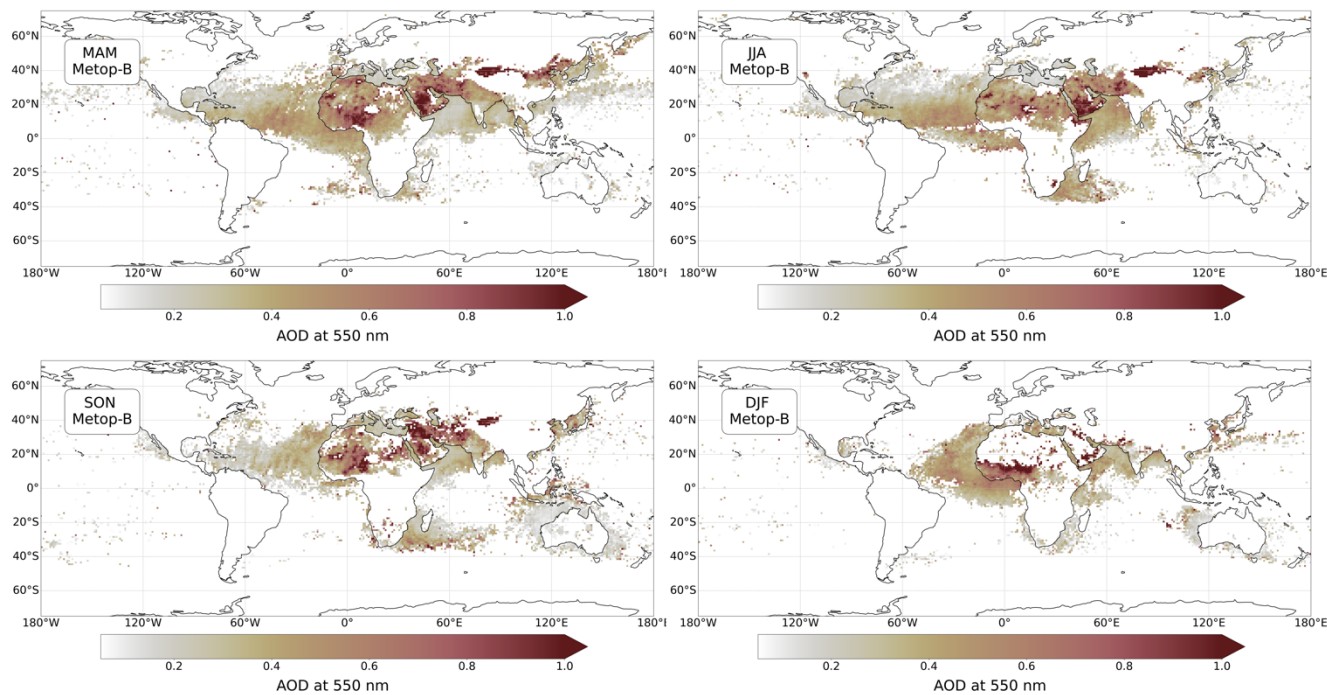

**Figure 16. Seasonal distributions for dust-related PMAp total AOD for 2015 from the Metop-B CDR. The seasonal mean includes
only grid cells where the number of observations > 10.**




Figure 16 illustrates the seasonal variation of dust-related total AODs from Metop-B for 2015. Highest dust related AODs are observed at the source areas (AOD > 0.6), while at the transport areas the mean AOD values are typically lower than 0.4. The Saharan outflow region demonstrates relatively consistent sampling throughout the year, while other areas, including Saharan Africa, show a lack of valid AOD observations during the Northern Hemisphere winter (DJF). This paucity of observations is partly due to the mapping restriction requiring at least 10 valid dust-related AOD observations per season for representation. In addition, snow cover extends to Northern Asia during these months that limit the AOD retrievals. However, even without the valid observation count restriction, these regions would still have a significant observation deficit.

The seasonal patterns for Metop-B in Figure 16 are comparable to Metop-A, although the overall number of Metop-A observations is generally somewhat lower due to the satellite scan angle limitation. Both PMAp CDRs capture the expected dust seasonality rather well, e.g. such as Sahara-to-Caribbean dust transport during NH spring and summer. On the other hand, the Taklamakan Desert exhibits consistently high average AODs across three seasons, aligning with AERONET validations that indicate a systematic overestimation by PMAp. This issue warrants further investigation.

When analysing closer the seasonal variation of dust-related AOD in specific study areas, the overall availability of observations needs to be considered. An example of this is illustrated in Figure 17, where monthly distributions for dust-related AODs are shown, defined from the whole Metop-B CDR. In the Saharan outflow area, it is seen how there is a clear increase in overall dust-related AOD observations during June and July, while the distribution also shifts towards larger values increasing the seasonal dust-related AOD median. In the Saharan overflow area, there is also a continuous dust-related AOD distribution present during the NH winter months, even though the number of observations is significantly lower.

In Saharan Africa, monthly dust-related total AOD distributions exhibit expected seasonal variability, with increased dust activity during spring and summer, as well as a lengthening tail toward higher AOD values, indicative of episodic dust events. However, the median of these distributions does not follow the expected pattern due to several factors. During winter, the distributions are discrete and highly variable, with some grid cells showing anomalously high AOD values likely caused by biases in the data. This leads to an unexpected increase in the median, even though winter is typically a low-dust season. In contrast, during summer, the dust-related AOD distributions extend continuously from low AOD values (<0.2) to higher values. However, the inclusion of these lower AOD values reduces the median, despite the notable increase in dust episodes during this period. This highlights that the median (or mean), while a useful statistical measure, may not always provide a complete picture, especially when distributions are irregular or skewed.





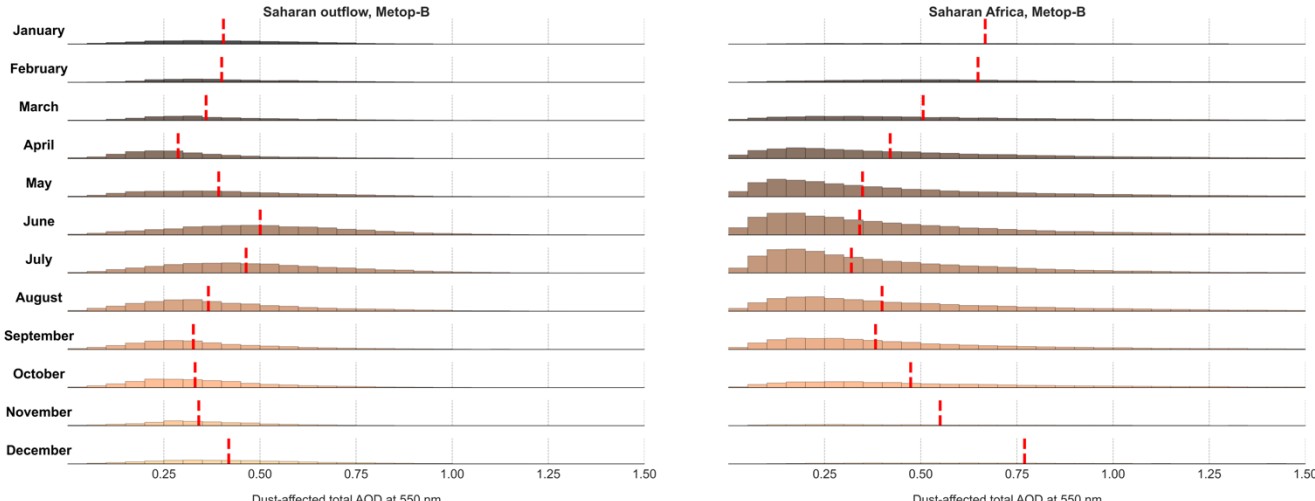

**Figure 17. Seasonal dust-related AOD distributions for the whole PMAp Metop-B CDR (2013-2019). Left panel represents the Saharan outflow area, while right panel represents the Saharan Africa. Red dashed lines are monthly distribution medians.**

**6 Summary and conclusions**

This study presents an evaluation of dust aerosols within the PMAp Climate Data Record (CDR) for Metop-A and -B. The analysis is divided into two parts: first, examining the occurrence of dust through spatial and temporal variations in dust-dominated total AODs across both CDRs; and second, validating the dust-dominated AODs using comparisons with AERONET observations. For AERONET comparisons the PMAp dust-dominated total AODs are extracted based on the retrieval classification of aerosol types. An important point to consider is the distinction between PMAp dust-dominated total AOD and the dust AOD (DAOD) provided by other satellite products. Unlike DAOD, which represents AOD composed exclusively of mineral particles, the PMAp dust-dominated AOD reflects the total AOD primarily influenced by (coarse) dust particles but still potentially including contributions from other aerosol types. Before the analysis both PMAp CDRs were filtered to reduce the effects of potential cloud-related biases. It is important to note that these thresholds were determined using data from the dust-impacted AERONET stations included in this study. If more stations were included, particularly those in non-dusty areas, different thresholds might be required.

Analysis of dust occurrence in the PMAp CDRs showed strong consistency between Metop-A and Metop-B. Both CDRs accurately captured the expected seasonal patterns of dust occurrence in key emission regions, including Saharan Africa, the Middle East, and Asia. Additionally, comparison with AERONET data confirmed that pixels identified as dust by PMAp were





consistently classified as 'dust' according to AERONET metrics. While some pixels meeting AERONET dust criteria were not classified as dust by PMAp, the overall results indicate that the dust detection performs reliably.


Comparison with AERONET data revealed that the correlations for both total PMAp AOD and dust-dominated total AOD ranged from moderate to good (R = 0.4-0.7), with relatively high mean biases and RMSEs. The RMSEs were typically between 50% and 80% of the mean AERONET AOD at the stations, with slightly higher values for dust-dominated AODs, suggesting that discrepancies in PMAp data were somewhat more pronounced for dust pixels. However, dust usually occurs in very

challenging retrieval conditions including bright surface, so to some extent elevated RMSEs are expected. Introducing an additional dust filter for AERONET AOD data did not significantly alter the statistical metrics for PMAp dust-dominated total AOD. Overall, the AERONET comparisons showed little systematic behaviour across the stations. A common feature was the high variability in the dust-dominated PMAp AOD bias. No significant differences were observed between the two CDRs.

For global-scale analysis of dust-related AODs, both CDRs were gridded into a 1° × 1° grid. The large-scale seasonal variations of dust-related AOD over source and transport regions were well captured by both Metop-A and Metop-B. However, it is important to note that sampling density can vary significantly across different study areas and seasons—for example, winter data may consist of only a few discrete observations, while summer provides a more complete distribution. Therefore, it is recommended to analyze the full distributions alongside median and mean values to ensure accurate conclusions.


The analysis also identified some challenges in the PMAp CDRs that warrant further investigation. AERONET comparisons in Central and Northern Asia (Asia 2) revealed a consistent overestimation of both total and dust-dominated PMAp AODs for both CDRs. Additionally, all study areas exhibited occasional outliers with unrealistically high dust-dominated AODs in both CDRs, but no single cause for these outliers was identified in this study. Over ocean, some of the scatter could be reduced by

applying the cloud filtering also for the AVHRR cloud fractions. Other potential causes for these outliers include e.g. fluctuations in surface data over bright surfaces. Furthermore, improvements in dust AOD retrievals could be achieved by upgrading the current dust aerosol model to better account for the non-sphericity of mineral dust particles.




## 7. Appendices

### Appendix A: Cloud filtering

Before conducting the analysis, both CDRs were examined for potential cloud contamination effects. A cloud correction (decontamination) scheme, implemented in the PMAp retrieval algorithm (Grzegorski et al., 2022), minimizes the impact of
residual clouds on aerosol property retrieval in partially cloudy scenes. However, this cloud correction factor is not fully applied to scenes flagged as dust. As a result, fractional cloud cover may still be present in PMAp dust scenes, even when the data is collocated with strictly cloud-free AERONET observations. Figure A1 presents the distribution of PMAp cloud fractions for Metop-A and Metop-B dust-dominated AODs collocated with AERONET data, combining all AERONET stations. While most observations occur under cloud-free conditions, there are instances where fractional cloud cover is
present.

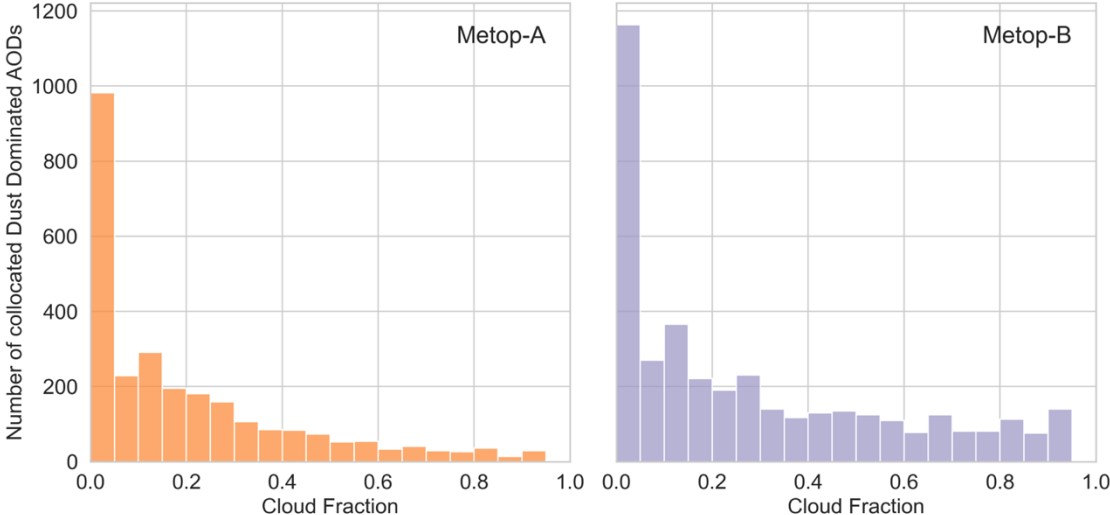

**Figure A1 Distribution of PMAp cloud fractions for PMAp dust-dominated AOD observations that have been collocated with AERONET observations. Left panel shows the distribution for METOP-A and right panel for Metop-B. For these distributions collocated AERONET observations for all stations considered in this study have been merged.**






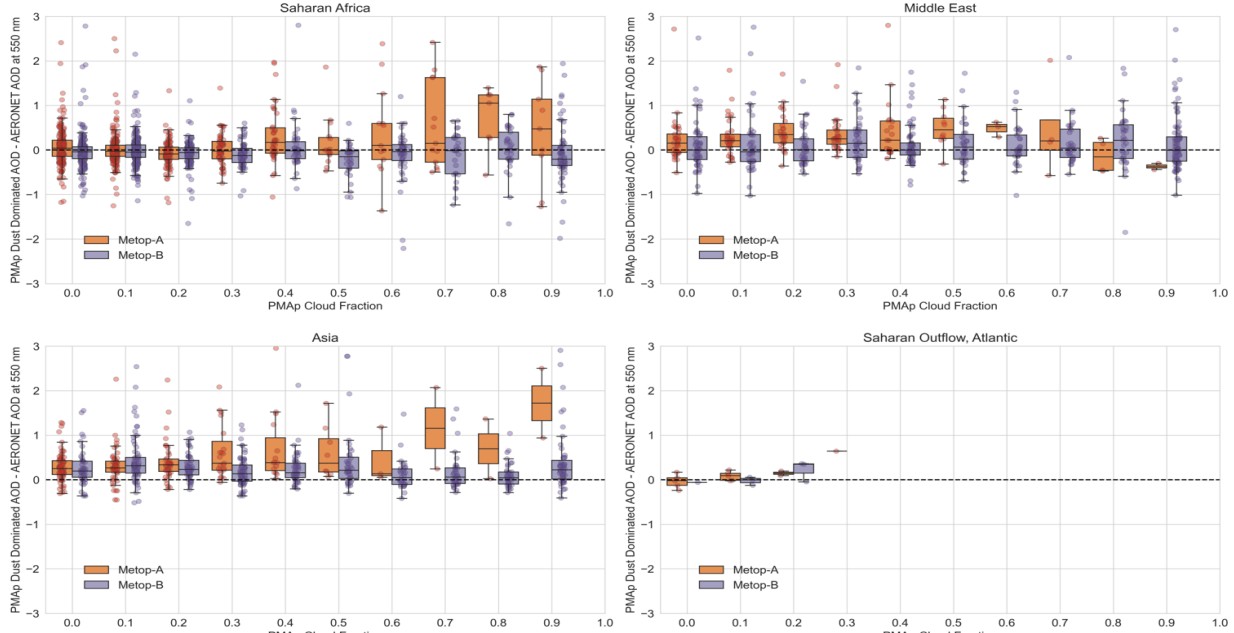

**Figure A2. PMAp dust-dominated AOD bias as a function of PMAp cloud fraction at the four main study areas.**

Figure A2 illustrates the PMAp dust AOD bias as a function of PMAp cloud fraction across the four main study areas. Notably,

relatively large cloud fractions are observed predominantly in regions where PMAp retrievals occur over bright land surfaces. The AOD bias for dust-dominated scenes appears to be distributed across all cloud fraction bins, suggesting that cloud fraction alone cannot explain the variability in AOD bias. However, lower cloud fractions tend to show a more concentrated distribution of AOD bias around smaller absolute values, whereas higher cloud fractions are associated with a wider dispersion of AOD bias. This pattern suggests that while cloud fraction is not the sole contributor to the observed bias, increasing cloud cover—

particularly over bright land surfaces—may lead to more pronounced discrepancies in AOD retrievals, potentially due to residual cloud contamination in dust scenes. Based on this analysis the threshold for PMAp cloud fraction was set to 0.4, that was applied for consistency for both CDRs.

For dust scenes over ocean an additional cloud filter was applied, based on the AVHRR cloud fraction. Based on the tests

carried out at Capo Verde station, the threshold was set to 0.6. Figure A3 shows the PMAp dust-dominated AOD bias as a function of AVHRR cloud fraction.





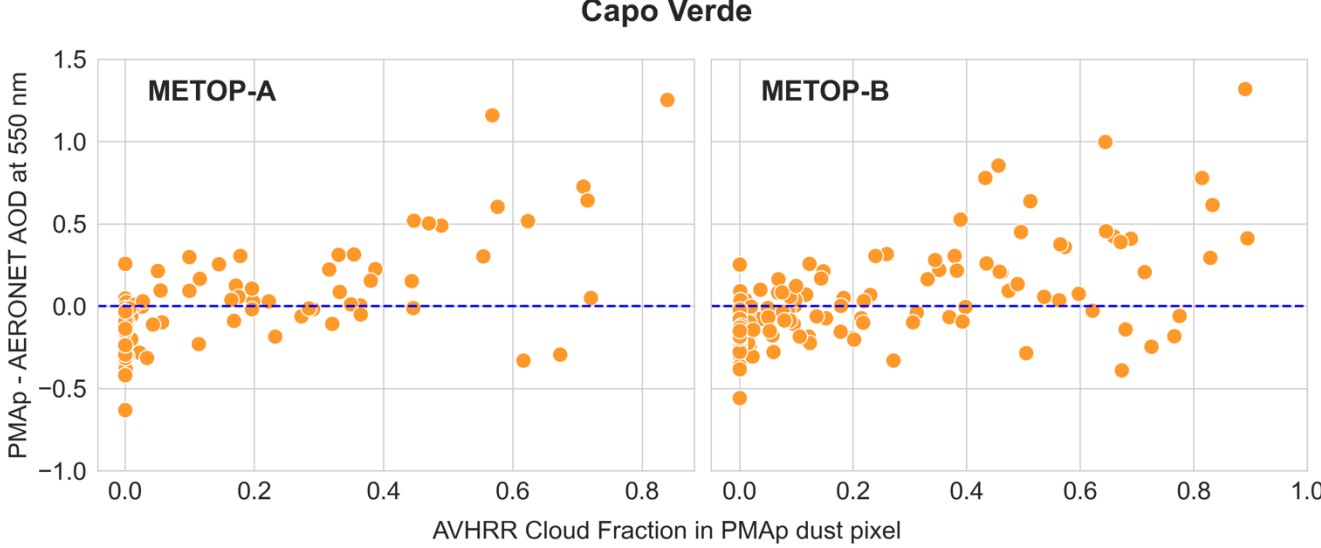


**Figure A3.** PMAp dust-dominated AOD bias as a function of AVHRR cloud fraction over ocean pixels.



**Appendix B: Supplementary figures for PMAp validation against AERONET**

**Saharan Africa**

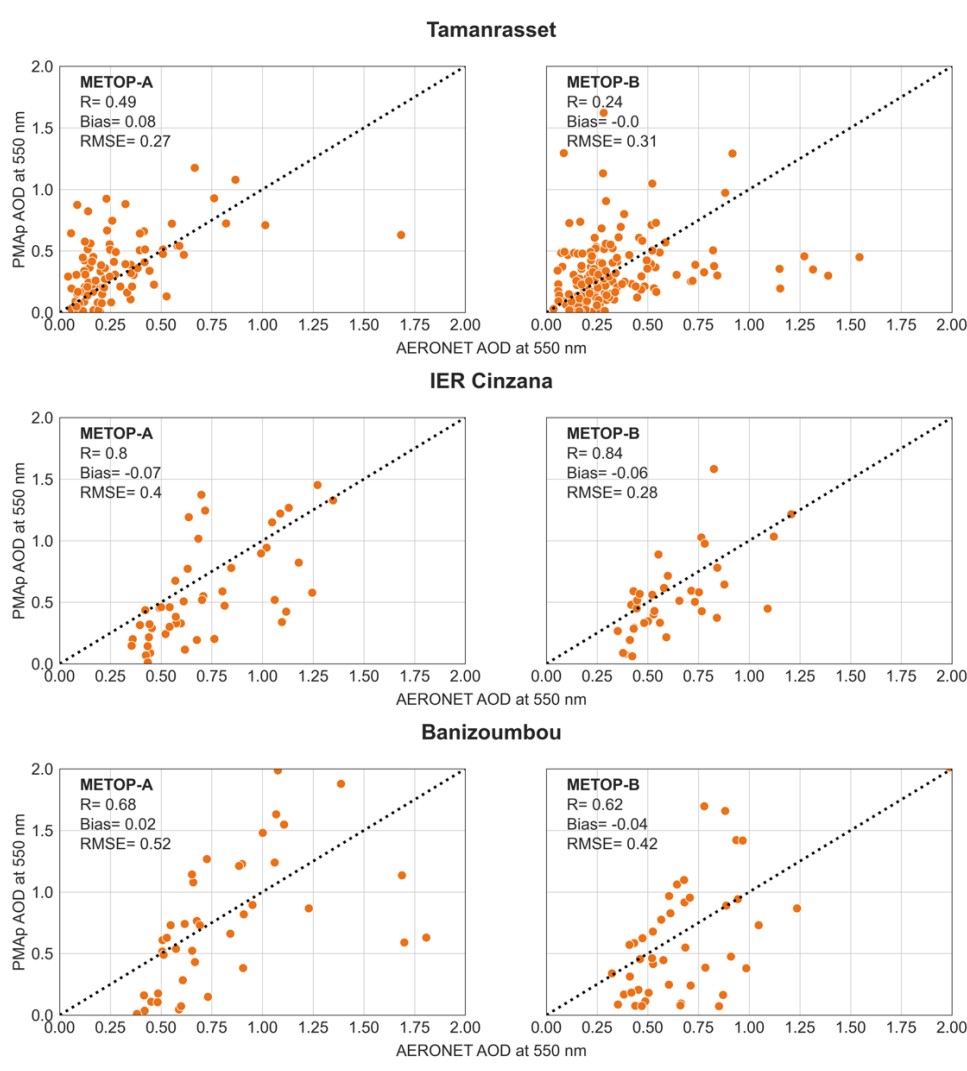


**Figure B1. Comparison of PMAp dust-dominated total AOD against dust-filtered AERONET data. Dashed black line indicate 1:1 line.**






**Figure B2. PMAp dust-dominated total AOD bias vs. AERONET total AOD at Tamanrasset and IER Cinzana. Grey points show individual bias values, while blue and red lines indicate the bin medians with standard deviations. Bins contain an equal number of observations.**



**Middle East**

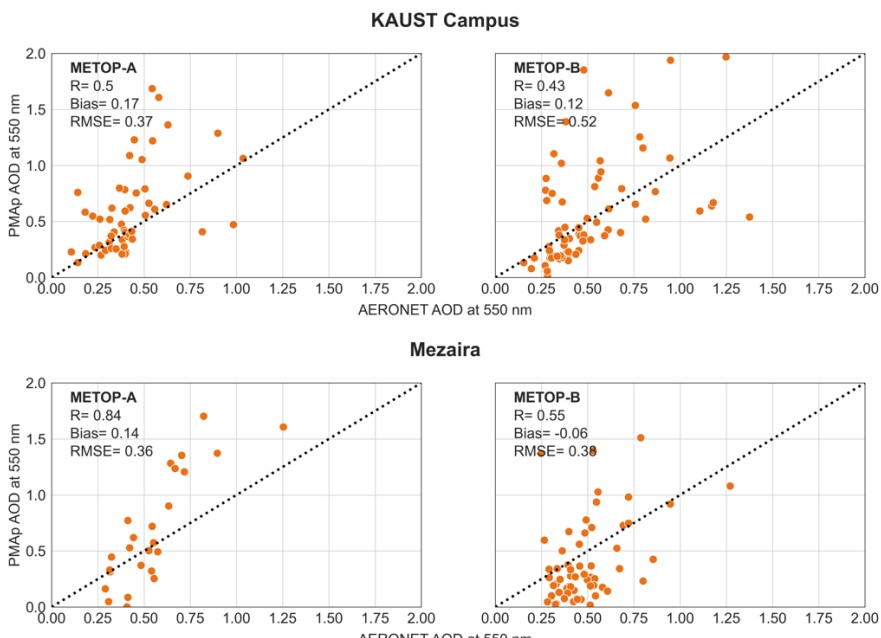

**Figure B3. Comparison of PMAp dust-dominated total AOD against dust-filtered AERONET data. Dashed black line indicate 1:1 line.**

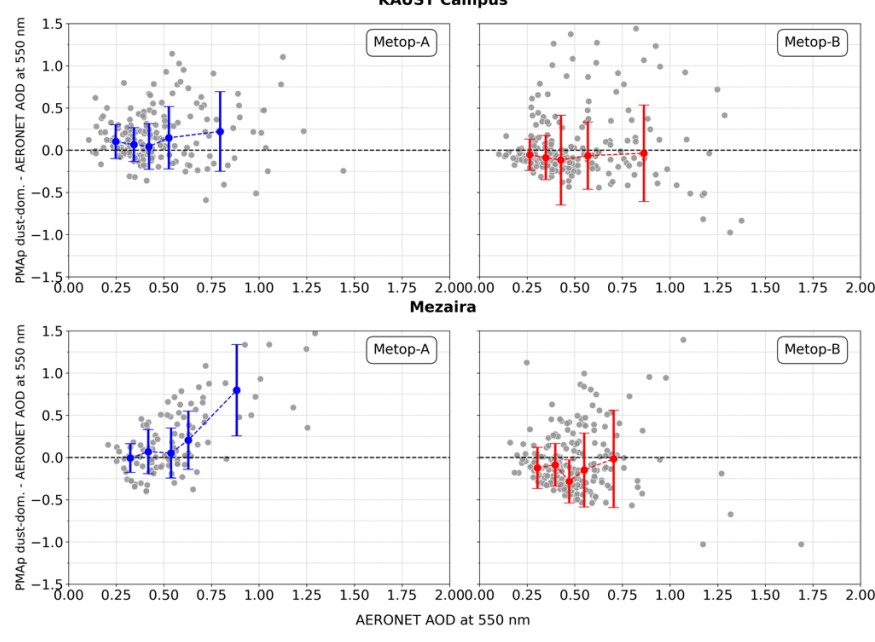


**Figure B4. PMAp dust-dominated total AOD bias vs. AERONET total AOD at Middle East stations.**





**Asia 1**

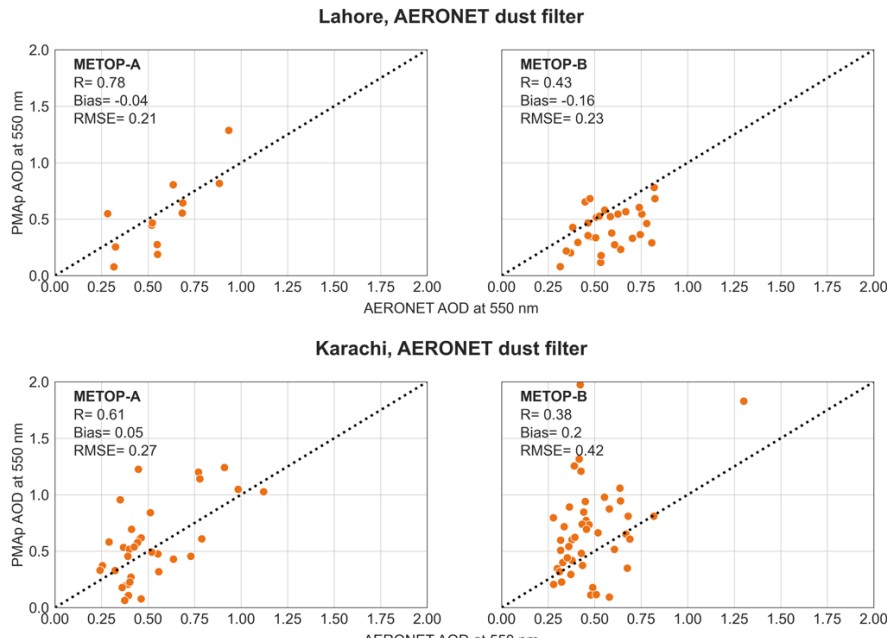

**Figure B5. Comparison of PMAp dust-dominated total AOD against dust-filtered AERONET data. Dashed black line indicate 1:1 line.**

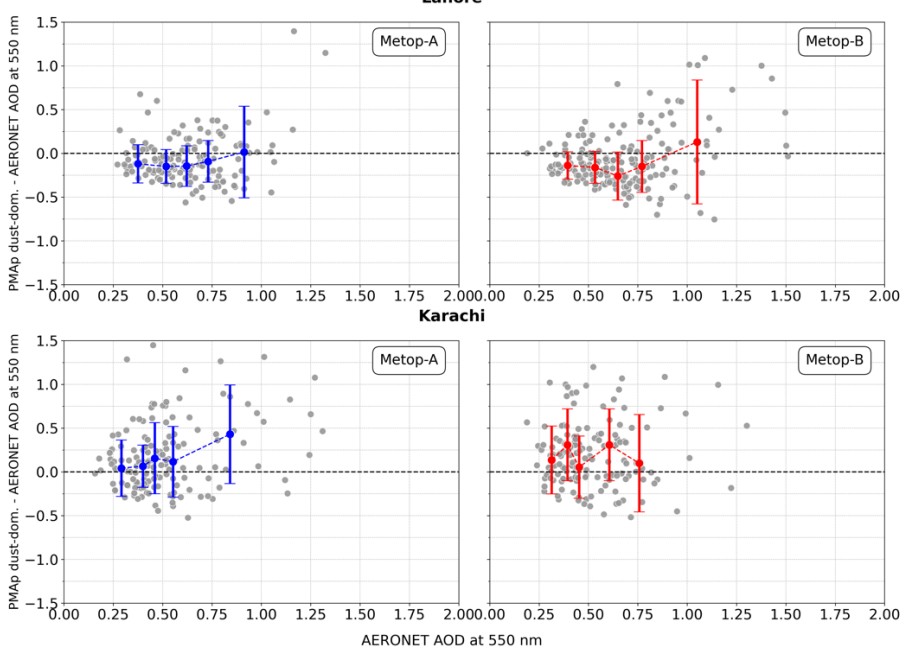

**Figure B6. PMAp dust-dominated total AOD bias vs. (unfiltered) AERONET AOD at Asia 1 stations.**



**A2.4 Asia 2**

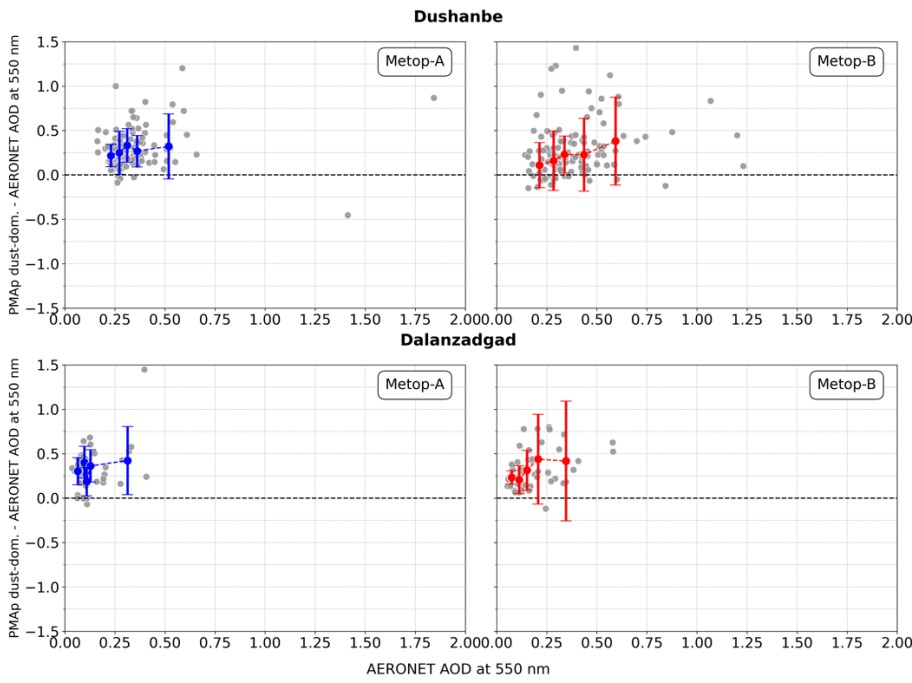


**Figure B7. Comparison of PMAp dust-dominated total AOD against dust-filtered AERONET data. Dashed black line indicate 1:1 line. For Dalanzadgad there were not enough collocated observations to carry out analysis.**


**Figure B8. PMAp dust-dominated total AOD bias vs. AERONET total AOD at Asia 2 stations.**






## A2.5 Saharan outflow

### Capo Verde

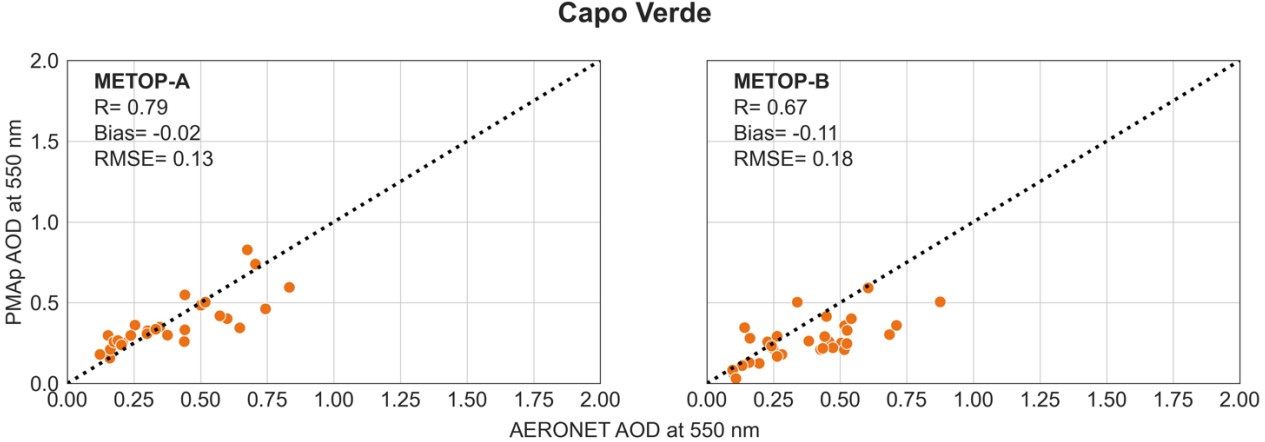

**Figure B9.** Comparison of PMAp dust-dominated total AOD against dust-filtered AERONET data. Dashed black line indicate 1:1

### Capo Verde

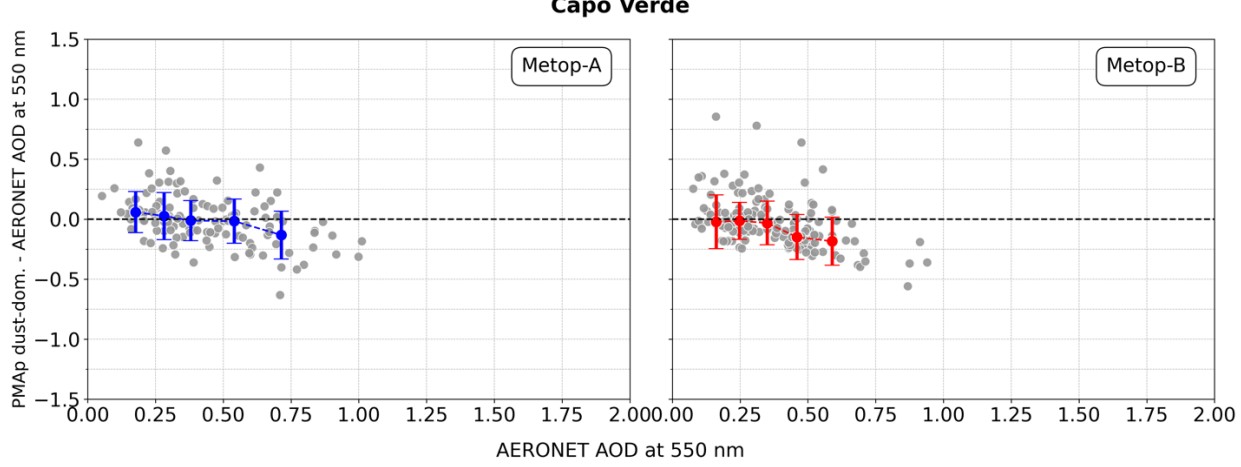


**Figure B10. PMAp dust-dominated total AOD bias vs. AERONET total AOD.**





## 8. Data and code availability

The PMAp CDR data are available via EUMETSAT Data Store or WEkEO at
https://user.eumetsat.int/catalogue/EO:EUM:DAT:0579/access. The AERONET data are available from NASA Goddard
Space Flight Center at https://aeronet.gsfc. nasa.gov/new_web/data.html.  Gridded Level 3 PMAp CDR datasets created in
this work and related codes are available upon request from the first author.

## 9. Author Contribution

AMS, MDB, SJ, and FF conceptualized the study and developed the methodology. SM and NF provided the platform for data processing.
AMS, MDB, DLC, SJ, and FF contributed to the data analysis. All contributed to an internal review of the manuscript.

## 10.  Competing interests

The authors declare that they have no conflict of interest.

## 11. Acknowledgements

We thank the AERONET PIs, Co-PIs and their staff for establishing and maintaining the sites used in this investigation.

## 12. Financial Support

This work is supported by EUMETSAT Service for User Guidance on Copernicus Atmospheric Composition Satellite Datasets
project (EUMETSAT Contract EUM/CO/22/4600002691/FFi).

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
