# Peer review of "Evaluation of the dust-dominated total AOD extracted from the PMAp satellite Climate Data Record"

_EGUsphere, 2025_

## Author Comment (AC1)

**Review #1**

Author's response to the review of:

**Evaluation of the dust-dominated total AOD extracted from the PMAp satellite Climate Data Record**

*Anu-Maija Sundström, Marie Doutriaux-Boucher, Soheila Jafariserajehlou, Dominika Leskow-Czyzewska , Simone Mantovan , Noemi Fazzini, Bertrand Fougnie, and Federico Fierli*

*Submitted to Atmospheric Measurement Techniques*

In the following, **R1** denotes the reviewer's comments and **A** denotes the authors' responses.

**General Comments**

**R1:** This manuscript presents a comprehensive validation of the dust-dominated total AOD derived from the PMAp CDR, with a particular focus on dust detection and AOD retrieval accuracy. The study uses high-quality AERONET ground-based observations across multiple global dust hotspots and applies robust comparison techniques, including both general and dust-specific evaluation metrics. The analysis is detailed, and the case study coverage is broad, making this an important contribution to satellite aerosol product validation literature. In addition, the manuscript is well-structured and the goals are clearly stated. The validation strategy is sound, and the distinction between "dust-dominated" AOD and conventional DAOD is clearly explained. The study highlights strengths and limitations of the PMAp dust product and provides guidance for future product use and development. I recommend the paper for publication after minor revisions to address the comments below.

*A: We sincerely thank Referee 1 for the thorough review and very positive feedback on our manuscript, and we greatly appreciate the recognition of the value of our work. The encouraging comments and constructive suggestions are very helpful, and we have carefully considered them in our revisions to further improve the manuscript.*

**Scientific/Technical Comments**

**R1-1:** line-594, section 5 on definition and Interpretation of Dust-Related AOD.

The definition of "dust-related AOD" as the mean AOD within grid cells where any dust detection occurred, may blur interpretation in regions with aerosol mixtures (e.g., Sahel during biomass burning). Please expand the discussion on the implications of this choice: how might this affect conclusions from Figure 16 and the seasonal diagnostics? Could a sensitivity test (e.g., thresholding dust pixel fraction) be included or at least recommended?

**A-1:** *We agree that the interpretation of the gridded "dust-affected AOD" is challenging, which is ultimately related to how the dust-dominated AOD should be interpreted in PMAp. We chose not to use any threshold for "dust-labelled pixels" in the gridding, because with PMAp data it is not possible to quantify the dust fraction to the total AOD. Even if thresholds would be introduced, the PMAp observations are always a mixture of several aerosol types, even if one type is assumed to dominate. We have added some clarifications on this to section 5 and Discussion.*

**R1-2:** line-581, section 4.4 the Central Asia region shows a consistent positive bias. The authors could elaborate further: are there known limitations in the GOME-2 GLER surface reflectance climatology for this region? Could dust mineralogy or vertical structure differ significantly in a way that violates retrieval assumptions (e.g., stronger absorption)? In addition, several regions exhibit sporadic high-AOD outliers. Even a brief diagnostic on a few cases (e.g., examining viewing angle, surface reflectance, or retrieval geometry) could help determine whether these are algorithm edge cases. This would provide useful guidance for future product refinement.

**A-2:** *We conducted a systematic investigation of high-AOD outliers using all variables available in the PMAp CDRs, including satellite–sun viewing geometry, sensor scan angle, cloud fraction, land–sea fraction, retrieval diagnostics (e.g., reflectance inhomogeneity), and distance to the AERONET site. Outliers occur more frequently over land than over ocean but otherwise we found no robust indicator or threshold among the available parameters in the CDR auxiliary data that would reliably flag and remove them. The CDRs do not include key land-surface descriptors (e.g., GLER), which would be valuable for diagnosing these cases. Because our analysis is limited to the parameters stored in the CDR files, identifying root causes will require retrieval-level studies with additional ancillary information, e.g. related to surface characteristics.*

*Central Asia is the only region in our study with seasonal snow, which may introduce residual artifacts despite the CDR snow/ice mask. The complex topography around Dushanbe (surrounded by mountains) adds further challenges, especially given the coarse GOME-2 pixel size. We would think that the surface-related reasons could be more important here than the dust mineralogy in explaining the systematic bias. However, again because our analysis is limited to the variables available in the CDRs, fully diagnosing the causes of the systematic overestimation will require retrieval-level experiments and another dedicated study.*

*These caveats are briefly noted in the Discussion. A sentence about complex topography and seasonal snow has been added to the beginning of chapter 4.3.4.*

**R1-3:** Line 374: The statement that AERONET AOD distributions for PMAp "dust" and "other" are clearly distinct in the Saharan outflow seems overstated. The figure shows substantial overlap, please qualify this interpretation.

**A-3:** *We have modified the text in line 384, regarding Saharan outflow as follows: "....Saharan outflow regions, the differences between the two PMAp categories is somewhat more*

*apparent, but still with substantial overlap.". (In line 374 there was no mention of Saharan outflow).*

**R1-4:** Line 398: Clarify which comparison approach corresponds to which figure. Explicitly stating that Figure 5 corresponds to comparisons 1 and 2, and that Figure A2.1 represents comparison 3, would improve reader orientation.

**A-4:** *The text has been modified by adding references to concerning figures, as suggested.*

**R1-5:** line-301, the threshold (α ≤ 0.75) is used to identify coarse-mode dominance. Please clarify whether this is based on literature or derived from your Figure 4. If the latter, a forward reference would help readers understand the justification.

**A-5:** *The threshold has been derived from literature, based on method by Gkikas et al., 2021. The reference is given on line 298. The original text has been modified by adding "thresholds" for clarification as follows: "Following the method and thresholds outlined by Gkikas et al. (2021), …."*

**Other minor comments**

**R1-6:** Line 104: "Aerosol Optical Depth (AOD)" was already defined; abbreviation does not need repeating.

**A-6:** *Aerosol Optical Depth removed.*

**R1-7:** Line 181: Only introduce full name of AERONET at first mention.

**A-7:** *Full name removed from line 181.*

**R1-8:** Line 205: Add degree symbols (°) to latitude and longitude coordinates.

**A-8:** *Symbols added.*

**R1-9:** Line 300: Only define "Single Scattering Albedo (SSA)" at first use.

**A-9:** *Corrected.*

---

## Author Comment (AC2)

**Review #2**

Author's response to the review of:

**Evaluation of the dust-dominated total AOD extracted from the PMAp satellite Climate Data Record**

*Anu-Maija Sundström, Marie Doutriaux-Boucher, Soheila Jafariserajehlou, Dominika Leskow-Czyzewska, Simone Mantovani, Noemi Fazzini, Bertrand Fougnie, and Federico Fierli*

*Submitted to Atmospheric Measurement Techniques*

In the following, **R2** denotes the reviewer's comments and **A** denotes the authors' responses.

**General comments**

**R2:** In general, this paper combines multiple input datasets to create a spatial and temporal description of "DAOD". The resulting data exhibit some inherent weaknesses, although they provide valuable insights into large-scale phenomena and dust occurrences. The methodology and the quality of the results have the potential to improve, particularly with advances in satellite instrumentation and the expansion of ground-based observations.

The author clearly describes both the strengths and weaknesses of the methodology. The selected ground sites correspond to different local conditions, providing an opportunity for a robust evaluation of the PMAp product.

**A:** *We thank the reviewer for the constructive and thoughtful evaluation of our manuscript. We appreciate the acknowledgment of both the strengths and limitations of our approach. This feedback has been very helpful in guiding our revisions and improving the manuscript. Our detailed responses to the comments are provided below.*

**Suggestions for Minor Improvements**

**R2-1:** A further analysis of seasonal patterns would be valuable—both within the final product itself and in comparison to AERONET or other established dust products.

**A-1:** *The seasonal behaviour is examined both in the AERONET comparisons (time series) and, globally, in Sect. 5. For co-located AERONET–PMAp pairs, the dust-dominated PMAp AOD follows the AERONET seasonal cycle (even though the plot is not specifically shown), and the shapes are broadly consistent. Seasonal cycles derived from AERONET alone can differ from PMAp, however, largely because of PMAp uneven temporal sampling (e.g., sparse winter observations). Sampling effects and their implications are discussed in Sect. 5, where PMAp seasonal variability is shown and discussed over broader dust source and transport regions.*

**R2-2:** It would be desirable for the authors to quantify the extent of missing data caused by systematic limitations in satellite observations. This would help demonstrate the impact of such limitations on the final results and could highlight potential constraints for the broader application of the method.

**A-2:** *It is difficult to provide a single definitive estimate because data gaps depend on local conditions: variation of cloudiness, solar zenith angle, seasonal snow cover, and other factors. As an illustration (Fig. 17), over Saharan Africa the number of valid observations in summer (May–August) is more than ten times that in winter (December–January). Over the Atlantic outflow, the seasonal contrast is smaller but still substantial: summer yields roughly four times as many observations as winter. As discussed in Sect. 5 and the Discussion, we recommend analysing the full AOD distributions in parallel with seasonal means or medians to quantify how gaps in sampling (missing data) could influence the inferred patterns.*

**Technical Recommendations and Errors**

**R2-3:** Due to the very dense data shown in most of the figures, I recommend using vector graphics formats (e.g., PDF, SVG, EPS) rather than raster formats like PNG or JPEG. This would ensure that the figures maintain full clarity when the article is viewed electronically—particularly for map-based figures such as Figures 1, 3, and 16.

**A-3:** *The map-based figures have been converted into pdf-format.*

**R2-4:** In line 430, the text references Figure A2.2, but the actual figure is labelled A2. Additionally, the font size in Figure A2 should be increased for better readability.

**A-4:** *The labelling has been corrected, and font size in the figure has been increased.*

**R2-5:** Figures A2.3, A2.4, and A2.10 are mis referenced similarly to A2.2 and should be corrected.

**A-5:** *Mis-references have been corrected.*

**R2-6:** Section numbering in the appendix is inconsistent. For example, on page 39, the section is titled "A2.4 Asia 2", whereas on page 38 it is simply "Asia 1". This should be standardized.

**A-6:** *Corrected.*

**R2-7:** In Figure 15, to improve readability, I suggest using colour scales with continuous transitions. For example:
- When the scale includes negative values, use a **blue–white–red** gradient, with white centered at zero.
- When the scale is non-negative, use a gradient such as **green–red** or **white–red**.

**A-7:** *We have changed the color scales as continuous, as suggested, and modified the color palettes.*